# TDM-R1: Reinforcing Few-Step Diffusion Models with Non-Differentiable Reward

Yihong Luo [1]    Tianyang Hu [2]    Weijian Luo [3]    Jing Tang [1 4]

## Abstract

While few-step generative models have enabled powerful image and video generation at significantly lower cost, generic reinforcement learning (RL) paradigms for few-step models remain an unsolved problem. Existing RL approaches for few-step diffusion models strongly rely on back-propagating through differentiable reward models, thereby excluding the majority of important real-world reward signals, e.g., non-differentiable rewards such as humans' binary likeness, object counts, etc. To properly incorporate non-differentiable rewards to improve few-step generative models, we introduce TDM-R1, a novel reinforcement learning paradigm built upon a leading few-step model, Trajectory Distribution Matching (TDM). TDM-R1 decouples the learning process into surrogate reward learning and generator learning. Furthermore, we developed practical methods to obtain per-step reward signals along the deterministic generation trajectory of TDM, resulting in a unified RL post-training method that significantly improves few-step models' ability with generic rewards. We conduct extensive experiments ranging from text-rendering, visual quality, and preference alignment. All results demonstrate that TDM-R1 is a powerful reinforcement learning paradigm for one- and few-step text-to-image models, scaling from 0.6B to 6B parameters while achieving state-of-the-art performance on both in-domain and out-of-domain metrics.

## 1. Introduction

Achieving rapid, high-fidelity image and video generation continues to be a fundamental objective in the field of Artificial Intelligence Generated Content (AIGC). Recent advancements in few-step generative models, e.g., diffusion distillation (Luo et al., 2023; 2024a; Yin et al., 2023; Zhou et al., 2024; Xie et al., 2024b; Luo et al., 2025c), have shown successes in ultra-fast photo-realistic images and videos generation with accelerations up to 50 times that of diffusion models. With ultra-fast generation efficiency and leading performances, few-step models are becoming a standard with large-scale serving in industry products (Cai et al., 2025; Labs, 2024). *Despite significant gains in speed and visual fidelity, few-step models still struggle with certain challenges, e.g., precise instruction following, complicated text rendering, and proper object positioning.*

In recent years, Reinforcement learning has shown significant ability in improving deep learning models' specialized abilities, ranging from large language models (Ziegler et al., 2019; Ouyang et al., 2022; Shao et al., 2024; DeepSeek-AI, 2025), standard diffusion models (Liu et al., 2025; Luo et al., 2025d), to few-step generative models (Luo, 2024; Luo et al., 2025a; Ren et al., 2024). It is expected that proper RL paradigms are capable of addressing the mentioned challenges of few-step image or video generation.

Though existing papers have explored RL methods for few-step generative models (Luo, 2024; Ren et al., 2024; Luo et al., 2025a), their algorithms rely on a narrow assumption:

- *they require reward signals to be differentiable, such that the outputs of the few-step generative models are able to backpropagate through the rewards.*

Such a narrow requirement clearly excludes a vast array of essentially non-differentiable reward signals, such as human binary preferences in real-world cases, discrete object countsat, and correctness of text-rendering via OCR models. On the other side, as pointed out by representative processes in LLM, e.g., Deepseek-R1 (DeepSeek-AI, 2025), rapid successes of RL in large language models clearly indicate the importance of generic non-differential rewards in unlocking models' hidden potentials.

To solve the key problem of *using non-differentiable rewards*, we introduce **TDM-R1**, a novel reinforcement learning paradigm designed to reinforce these few-step models using *free-form non-differentiable reward feedback* without requiring additional ground-truth image data. **TDM-R1** is

[1]Hong Kong University of Science and Technology [2]Chinese University of Hong Kong, Shenzhen [3]hi-lab, Xiaohongshu Inc [4]Hong Kong University of Science and Technology (Guangzhou). Correspondence to: Jing Tang <jingtang@ust.hk>.

*Proceedings of the 43rd International Conference on Machine Learning*, Seoul, South Korea. PMLR 306, 2026. Copyright 2026 by the author(s).

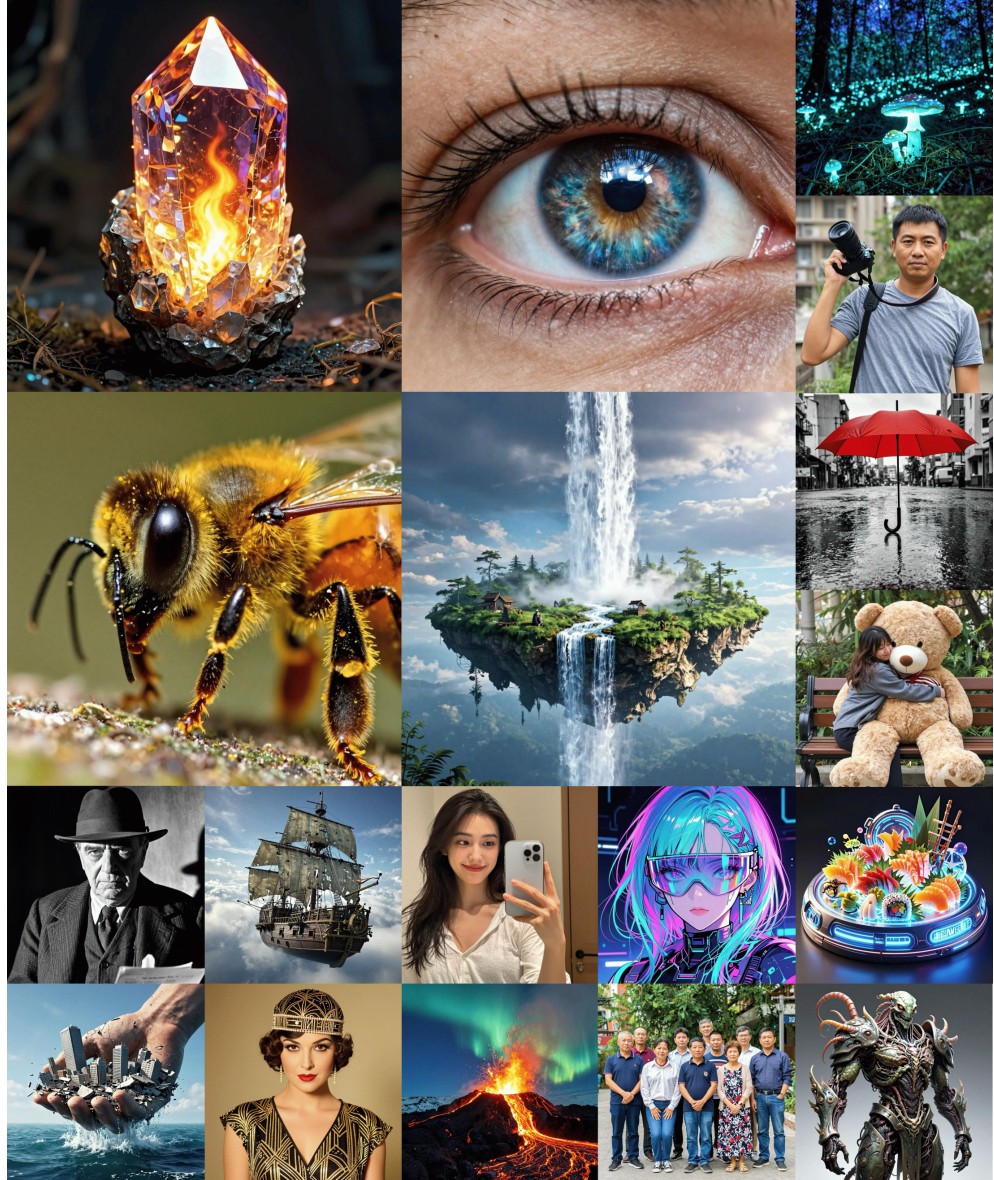

*Figure 1.* Samples generated by **TDM-R1** with 4NFE, reinforcing the 6B Z-Image.

built upon Trajectory Distribution Matching (TDM) (Luo et al., 2025c), a typical few-step generative model. The core idea of TDM-R1 is to utilize the deterministic trajectories of TDM to assign rewards to intermediate denoising steps via an efficient and unbiased reward estimate for every sample along the trajectory. We also propose a diffusion-parameterized dynamic surrogate reward trained using pairwise groups to provide stable RL supervision at each step.

By decoupling the learning process into surrogate reward learning and generator optimization, TDM-R1 enables few-step models to achieve performance levels that even surpass many-step counterparts. In Section 4, we conduct extensive experiments across compositional image generation, ranging from visual text rendering and human preference

alignment to validate the superiority of TDM-R1 on both in-domain and out-of-domain metrics. ***Remarkably, our method enables models to outperform expensive 80-NFE base models using only 4 NFE***. Most notably, on the rigorous GenEval benchmark (Ghosh et al., 2023), TDM-R1 boosts performance from 61% to 92%—significantly surpassing the 80-NFE base model (63%) and the commercial state-of-the-art GPT-4o (84%).

## 2. Preliminaries

**Diffusion Models.** Diffusion Models (DMs)(Sohl-Dickstein et al., 2015; Ho et al., 2020) establish a forward diffusion process that corrupts input sample $\mathbf{x}_0$ with Gaussian noise over $T$ discrete timesteps. This forward

A photo of burger **left of** a apple | A photo of **five** cute corgi | A photo of a green Pikachu | A photo of a cat **right of** a cute rat | A girl with black dress and red hair **right of** a boy with blue hair

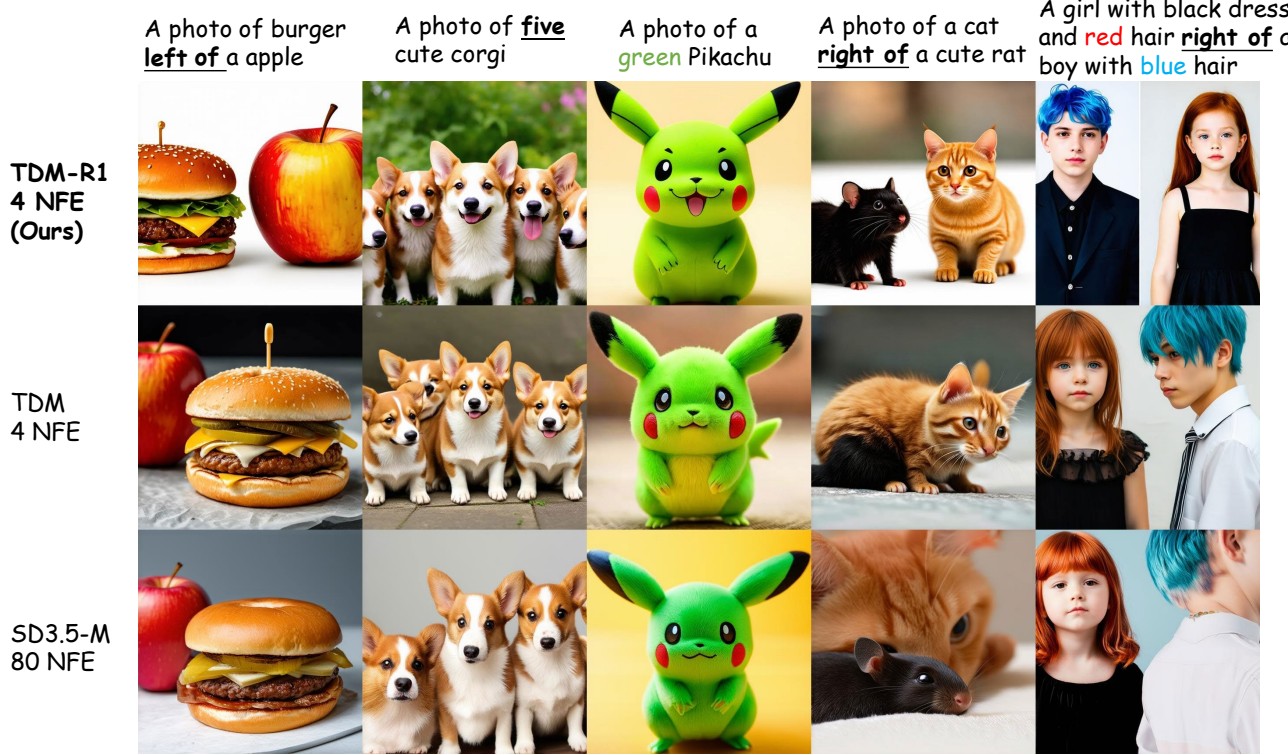

TDM-R1 4 NFE (Ours)

TDM 4 NFE

SD3.5-M 80 NFE

*Figure 2.* Qualitative comparisons of TDM-R1 with GenEval signals against competing methods. It can be seen that our TDM-R1 can accurately follow the instructions, while keeping a strong visual quality. The same initial noise is used to generate all images.

process is defined by: $q(\mathbf{x}_t|\mathbf{x}) \triangleq \mathcal{N}(\mathbf{x}_t; \alpha_t \mathbf{x}, \sigma_t^2 \mathbf{I})$, where $\alpha_t$ and $\sigma_t$ are hyperparameters regarding the noise schedule. At any given timestep, noisy samples are generated as $\mathbf{x}_t = \alpha_t \mathbf{x} + \sigma_t \epsilon$, with $\epsilon \sim \mathcal{N}(\mathbf{0}, \mathbf{I})$. The learned reverse diffusion process is formulated as $p_\psi(\mathbf{x}_{t-1}|\mathbf{x}_t) \triangleq \mathcal{N}(\mathbf{x}_{t-1}; \mu_\psi(\mathbf{x}_t, t), \eta_t^2 \mathbf{I})$. Training proceeds by optimizing the neural network $f_\psi$ through the denoising objective

$$\mathbb{E}_{\mathbf{x}, \epsilon, t} \lambda_t ||f_\psi(\mathbf{x}_t, t) - \mathbf{x}||_2^2 \qquad (1)$$

A well-trained diffusion model can estimate the score via: $\nabla_{\mathbf{x}_t} \log p_t(\mathbf{x}_t) \approx s_\psi(\mathbf{x}_t) = -\frac{\mathbf{x}_t - \alpha_t f_\psi(\mathbf{x}_t, t)}{\sigma_t^2}$.

**Trajectory Distribution Matching (TDM).** TDM (Luo et al., 2025c) aligns the student trajectory and teacher trajectory at the distributional level. The learning is performed by minimizing the integral reverse KL divergence (Luo et al., 2023; Wang et al., 2023) at each step of the $K$-step student trajectory as follows:

$$\nabla_\theta L(\theta) = \mathbb{E}_{k, t \geq t_k} \lambda_t [\nabla_{\mathbf{x}_t} \mathrm{KL}(p_{\theta,k}(\mathbf{x}_t)||p_\psi(\mathbf{x}_t))] \frac{\partial \mathbf{x}_{t_k}}{\partial \theta}$$
$$\approx \mathbb{E}_{k, t \geq t_k, q(\mathbf{x}_t|\mathbf{x}_{t_k})} \lambda_t [s_{\mathrm{fake}}(\mathbf{x}_t) - s_\psi(\mathbf{x}_t)] \frac{\partial \mathbf{x}_{t_k}}{\partial \theta}, \qquad (2)$$

where $t_k = \frac{T}{K} k$ by default, $T$ is the terminal timestep, $p_{\theta,k}(\mathbf{x}_t) \triangleq \int q(\mathbf{x}_t|\mathbf{x}_{t_k}) p_\theta(\mathbf{x}_{t_k}) d\mathbf{x}_{t_k}$, $p_\theta(\mathbf{x}_{t_k})$ is the distribution of student trajectory at timestep $t_k$, the $p_\psi(\mathbf{x}_t)$ is the pre-trained teacher DM, and a online fake score $s_{\mathrm{fake}}(\mathbf{x}_t)$ is

employed to approximate the score of student distribution.

**Reward Modeling and RLHF.** Consider ranked pairs derived from a condition $\mathbf{c}$ denoted $\mathbf{x}_0^w \succ \mathbf{x}_0^l | \mathbf{c}$, where $\mathbf{x}_0^w$ represents the preferred sample and $\mathbf{x}_0^l$ represents the less preferred sample. The Bradley-Terry (BT) model characterizes the pairwise preferences as:

$$p_{\mathrm{BT}}(\mathbf{x}_0^w \succ \mathbf{x}_0^l | \mathbf{c}) = \sigma(r(\mathbf{c}, \mathbf{x}_0^w) - r(\mathbf{c}, \mathbf{x}_0^l)) \qquad (3)$$

where $\sigma(\cdot)$ is the sigmoid function. A parameterized reward network $r_\phi$ can then be learned through maximum likelihood estimation: $\min_\phi -\log p_{\mathrm{BT}}(\mathbf{x}_0^w \succ \mathbf{x}_0^l | \mathbf{c})$.

The target of RLHF is to optimize a conditional generator $p_\theta(\mathbf{x}_0|\mathbf{c})$ such that it maximizes an certain reward $r_\phi(\mathbf{c}, \mathbf{x}_0)$ while remaining close to a reference distribution $p_{\mathrm{ref}}$ via:

$$\max_{p_\theta} \mathbb{E}_{\mathbf{c}, \mathbf{x}_0 \sim p_\theta(\mathbf{x}_0|\mathbf{c})} [r(\mathbf{c}, \mathbf{x}_0)] - \beta \mathrm{KL}[p_\theta(\mathbf{x}_0|\mathbf{c})|p_{\mathrm{ref}}(\mathbf{x}_0|\mathbf{c})]$$
$$(4)$$

Here, $\beta$ controls the strength of the regularization.

## 3. Method

**Problem Setup.** Let $p_\theta$ denote a pre-trained few-step model parameterized by $\theta$. We assume access to a dataset of conditions $\mathcal{D}_c = \{\mathbf{c}_i\}_{i=1}^N$ and a reward function $r(\cdot, \cdot) : \mathcal{X} \times \mathcal{C} \to \mathbb{R}$ that measures the quality of generated samples $\mathbf{x} \in \mathcal{X}$ with respect to a given condition $\mathbf{c} \in \mathcal{C}$. *Our goal is to reinforce the few-step model $p_\theta$ accord-*

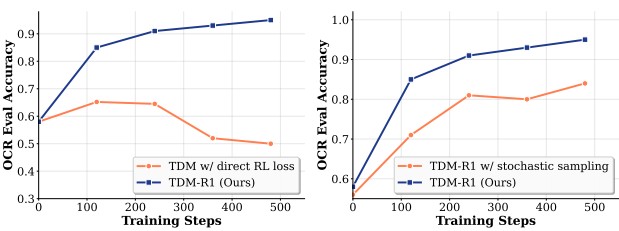

*Figure 3.* Compare the training performance and speed of TDM-R1 and potential baselines.

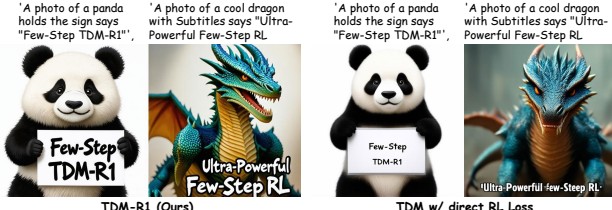

*Figure 4.* Compare TDM-R1 with the direct combination of TDM and RL loss.

*ing to the non-differentiable reward signal.* We adopt an online RL framework: at each training iteration, the current model $p_\theta$ produces a group of samples conditioned on $c \in \mathcal{D}_c$ through $K$-step sampling. The samples generated along each trajectory are collected into datasets $\mathcal{D}_k = \{(\mathcal{G}_i = \{\mathbf{x}_{t_k,j}\}_{j=1}^G, \mathbf{c}_i) \mid \mathbf{x}_{t_k,j} \sim p_\theta(\cdot, t_k|\mathbf{c}_i)\}$, which are subsequently scored by the reward function to obtain training signals. See Appendix. A for notation summary.

**Overview.** Our method addresses reinforcement learning of few-step diffusion models through three key components. First, in Section 3.1, we establish that deterministic sampling trajectories—as employed by trajectory distribution matching (TDM)—enable accurate reward estimation for intermediate steps (Equation (7)). Second, in Section 3.2, we address the incompatibility between standard diffusion RL methods and few-step generation by introducing a Surrogate Reward. This module learns a fine-grained, differentiable reward for each step along the trajectory via group-based preference optimization. Third, in Section 3.3, we formulate the learning objective for the few-step generator.

### 3.1. Accurate Intermediate Reward Estimation via Deterministic Trajectories

In order to leverage reward signals to reinforce a well-trained few-step model. An obvious challenge is that reward signals are generally defined over clean images, while few-step models perform inference by progressively denoising from noise to images, making it difficult to assign rewards to intermediate steps. Previous works often directly use the reward at the endpoint of the sampling trajectory as the reward for the entire diffusion path (Liu et al., 2025; Xue et al., 2025b), which might introduce high variance into the rewards assigned to intermediate steps. Fortunately, prior works (Luo et al., 2025b) have discovered that a conditional

probability over noisy images $p(\mathbf{c}|\mathbf{x}_t)$ can be obtained as follows:

$$p(\mathbf{c}|\mathbf{x}_t) = \int p(\mathbf{c}|\mathbf{x})p(\mathbf{x}|\mathbf{x}_t)dx \quad (5)$$

If we normalize $r(\mathbf{x}, \mathbf{c})$ to $[0, 1]$ and interpret this value as the probability that image $\mathbf{x}$ is a good sample given condition $\mathbf{c}$, i.e., $p(\text{"}\mathbf{x}\text{ is a good sample"}|\mathbf{x}, \mathbf{c}) = r(\mathbf{x}, \mathbf{c})$, we can model the reward for the noisy image $\mathbf{x}_t$ by:

$$r(\mathbf{x}_t, \mathbf{c}) = \int r(\mathbf{x}, \mathbf{c})p(\mathbf{x}|\mathbf{x}_t)dx = \mathbb{E}_{p(\mathbf{x}|\mathbf{x}_t)}r(\mathbf{x}) \quad (6)$$

This Equation (6) actually justifies the reasonableness of defining the reward for the entire diffusion path using the reward of $x_0$, which is a single-sample estimate of the Equation (6). The variance of this estimate depends on the variance of $p(\mathbf{x}|\mathbf{x}_t)$.

In other words, if $p(\mathbf{x}|\mathbf{x}_t)$ is a deterministic Dirac distribution—i.e., for diffusion, the path from $\mathbf{x}_t$ to $x_0$ is obtained via ODE sampling – then we can obtain an unbiased estimate of the reward for intermediate samples along the sampling path, which will significantly reduce the variance of reward estimation. This also explains why Mix-GRPO achieves superior performance by converting some SDE steps into ODE steps in RL's rollout (Li et al., 2026) — it reduces the variance of estimating rewards for intermediate samples along the trajectory.

Through the above analysis, we propose to conduct reinforcement learning based on TDM—a state-of-the-art few-step diffusion model that performs sampling through deterministic trajectories, allowing us to obtain accurate reward estimates for every intermediate sample along the trajectory, thereby achieving superior reinforcement post-training. Compared with the stochastic trajectories, the deterministic trajectories enable more accurate reward estimation for intermidate step, leading to faster convergence and better ultimate performance (Figure 3).

### 3.2. Surrogate Reward Learning

Given non-differentiable reward signals and a well-trained few-step diffusion model, a natural idea is to directly apply RL methods designed for standard diffusion models to few-step diffusion models. While this approach can leverage reward signals to some extent, it fails to sufficiently improve performance or generate high-quality images—outputs tend to be blurry, as shown in Figures 3 and 4. This is because RL methods for standard diffusion models that handle non-differentiable rewards can typically be reformulated as, or are equivalent to, weighted combinations of denoising losses (Wallace et al., 2024; Zheng et al., 2025; Xue et al., 2025a). Due to the inherent characteristics of denoising losses, they tend to produce blurry results with few sampling steps, making them incompatible with few-step diffusion models (Luo et al., 2025b).

To more effectively leverage non-differentiable reward signals, we propose learning a Surrogate Reward for reinforcement and using this Surrogate Reward to guide model learning. Inspired by the success of prior work (Wallace et al., 2024; Luo et al., 2025d), which have demonstrated the effectiveness of parameterizing reward models using the diffusion models, we adopt a similar approach. Specifically, they derive from the RLHF objective that the reward for a clean image $\mathbf{x}_0$ can be approximately parameterized as:

$$\tilde{r}_\phi(\mathbf{x}_0, \mathbf{c}) \approx \beta \mathbb{E}_{q(\mathbf{x}_{1:T}|\mathbf{x}_0)} \log \frac{p_\phi(\mathbf{x}_{0:T}|\mathbf{c})}{p_{\text{ref}}(\mathbf{x}_{0:T}|\mathbf{c})} + \beta \log Z(\mathbf{c})$$

Through a similar derivation, we can parameterize the surrogate reward for a noisy sample $\mathbf{x}_{t_k}$ as follows:

$$\tilde{r}_\phi(\mathbf{x}_{t_k}, \mathbf{c}) \approx \beta \mathbb{E}_{q(\mathbf{x}_{t_{k+1}:T}|\mathbf{x}_{t_k})} \log \frac{p_\phi(\mathbf{x}_{t_k:T}|\mathbf{c})}{p_{\text{ref}}(\mathbf{x}_{t_k:T}|\mathbf{c})} + \beta \log Z(\mathbf{c})$$
(7)

One could learn this Surrogate Reward through preference relations between pairwise positive and negative samples. However, this approach cannot leverage fine-grained rewards and relative preference relations within groups, limiting the effectiveness of reward learning. In contrast, group-based methods have achieved widespread success in both the language models and image generation (DeepSeek-AI, 2025; Liu et al., 2025; Luo et al., 2025d).

Therefore, we propose directly learning preference relations between pairwise positive and negative groups of noisy samples using the Bradley-Terry (BT) model. Following prior work (Luo et al., 2025d), we formulate the learning objective as follows:

$$\min - \log p(\mathcal{G}_k^+ \succ \mathcal{G}_k^- | \mathbf{c}) = -\log \frac{1}{1 + \exp(R(\mathcal{G}_k^-) - R(\mathcal{G}_k^+))}, \quad (8)$$

where $\mathcal{G}_k^+ \cup \mathcal{G}_k^- = \mathcal{G}_k$, and the group-level reward $R(\mathcal{G})$ is defined as the weighted sum of rewards for each sample within the group, i.e., $R(\mathcal{G}_k) = \sum_{\mathbf{x} \in \mathcal{G}_k} w(\mathbf{x}_{t_k}) \tilde{r}_\phi(\mathbf{x}_{t_k}, \mathbf{c})$.

The weight $\omega(\mathbf{x}_{t_k})$ controls the importance level of each noisy sample within the group, which is set to the absolute normalized value of the advantage within the group, i.e., $\omega(\mathbf{x}_{t_k}^i) = |A(\mathbf{x}_{t_k}^i)| = |\frac{r_i - \text{mean}(\{r_j\}_{j=1}^G)}{\text{std}(\{r_j\}_{j=1}^G)}|$. This design assigns larger weights to samples that are either significantly better or worse within the group, providing fine-grained learning signals.

The partition into positive and negative groups is based on the advantage $A(\mathbf{x}_{t_k}^i)$, i.e.,

$$\mathcal{G}_k^+ = \{\mathbf{x}_{t_k}^i : A(\mathbf{x}_{t_k}^i) > 0\}, \mathcal{G}_k^- = \{\mathbf{x}_{t_k}^i : A(\mathbf{x}_{t_k}^i) \leq 0\}$$

Through some calculation, a tractable upper-bound of loss

in Equation (8) can be rewritten in the following form:

$$L(\theta) \leq \mathbb{E}_{(\mathcal{G}_k^+, \mathcal{G}_k^-) \sim \mathcal{D}_k, k} \mathbb{E}_{t, q(\mathbf{x}_t|\mathbf{x}_{t_k})} \log \sigma(-\beta(T - t_k)\{$$
$$\sum_{\mathbf{x}_0 \in \mathcal{G}_k^+} w(\mathbf{x}_{t_k})[\text{KL}(q(\mathbf{x}_{t-1}|\mathbf{x}_t, \mathbf{x}_{t_k}) || p_\phi(\mathbf{x}_{t-1}|\mathbf{x}_t))$$
$$- \text{KL}(q(\mathbf{x}_{t-1}|\mathbf{x}_t, \mathbf{x}_{t_k}) || p_{\text{ref}}(\mathbf{x}_{t-1}|\mathbf{x}_t))] \quad (9)$$
$$- \sum_{\mathbf{x}_{t_k} \in \mathcal{G}_k^-} w(\mathbf{x}_{t_k}) \cdot [\text{KL}(q(\mathbf{x}_{t-1}|\mathbf{x}_t, \mathbf{x}_{t_k}) || p_\phi(\mathbf{x}_{t-1}|\mathbf{x}_t))$$
$$- \text{KL}(q(\mathbf{x}_{t-1}|\mathbf{x}_t, \mathbf{x}_{t_k}) || p_{\text{ref}}(\mathbf{x}_{t-1}|\mathbf{x}_t))]\}),$$

where we omitted the condition $\mathbf{c}$ for simplicity without loss of generality. We defer the derivations in the Appendix C. Under the Gaussian parameterization (Song et al., 2020), the posterior and model distributions are given by: $q(\mathbf{x}_{t-1}|\mathbf{x}_t, \mathbf{x}_{t_k}) \triangleq \mathcal{N}(\alpha_{t-1|t_k} \mathbf{x}_{t_k} + \sqrt{\sigma_{t-1|t_k}^2 - \eta_t^2}\epsilon, \eta_t^2 I)$ and $p_\phi(\mathbf{x}_{t-1}|\mathbf{x}_t) \triangleq \mathcal{N}(\alpha_{t-1} \frac{\mathbf{x}_t - \sigma_t \epsilon_\phi}{\alpha_t} + \sqrt{\sigma_{t-1}^2 - \eta_t^2}\epsilon_\phi, \eta_t^2 I)$.

**Dynamic Reference Model.** A frozen reference model may impose overly strong regularization, thereby hindering the learning of the reward model. Prior work has opted to update the reference model with theta at fixed intervals. However, this approach may introduce instability: $\theta$ might overfit to noisy signals at a certain step and subsequently be used as the reference model. To achieve better dynamic updating of the reference model, we propose to use an EMA (Exponential Moving Average) version of $\phi$ as the reference model. This not only relaxes the regularization to facilitate better reward learning, but also mitigates the risk of adopting a "bad" reference model that has overfitted to noisy signals.

### 3.3. Few-Step Generator Learning

We formulate the reinforcement learning training objective for the $K$-step generator in a form similar to standard RLHF (Luo et al., 2023; 2024b), consisting of reward maximization with a reverse KL regularization as follows:

$$L(\theta) = \mathbb{E}_{k, p_\theta(\mathbf{x}_{t_k})} - \tilde{r}_{\text{sg}(\phi)}(\mathbf{x}_{t_k}, \mathbf{c}) + \beta_g \text{KL}(p_{\theta,k}(\mathbf{x}_t) || p_\psi(\mathbf{x}_t)),$$
(10)

where $p_{\theta,k}(\mathbf{x}_t) \triangleq \int q(\mathbf{x}_t|\mathbf{x}_{t_k}) p_\theta(\mathbf{x}_{t_k}) d\mathbf{x}_{t_k}$. In the objectives described above, Surrogate Reward maximization provides a good learning signal to integrate non-differentiable reward feedback, while the marginal-level reverse KL regularization effectively grounds the generated samples to the base distribution parameterized by the pretrained diffusion model at the marginal distribution level. This differs from the KL regularization adopted in standard diffusion RL (Fan et al., 2024; Liu et al., 2025), which is essentially an instance-level constraint that requires each point along the trajectory to be consistent with the base model, leading to an unnecessarily difficult constraint.

Put the reward parameterization in Equation (7) into Equation (10), we can derive the learning gradient with some

*Table 1.* **GenEval Result.** We **highlight** the best scores among specific category. Obj.: Object; Attr.: Attribution.

| Model | Overall | Single Obj. | Two Obj. | Counting | Colors | Position | Attr. Binding |
|---|---|---|---|---|---|---|---|
| *Autoregressive Models:* | | | | | | | |
| Show-o (Xie et al., 2024a) | 0.53 | 0.95 | 0.52 | 0.49 | 0.82 | 0.11 | 0.28 |
| Emu3-Gen (Wang et al., 2024) | 0.54 | 0.98 | 0.71 | 0.34 | 0.81 | 0.17 | 0.21 |
| JanusFlow (Ma et al., 2025) | 0.63 | 0.97 | 0.59 | 0.45 | 0.83 | 0.53 | 0.42 |
| Janus-Pro-7B (Chen et al., 2025) | 0.80 | **0.99** | 0.89 | 0.59 | 0.90 | **0.79** | **0.66** |
| GPT-4o (Hurst et al., 2024) | **0.84** | **0.99** | **0.92** | **0.85** | **0.92** | 0.75 | 0.61 |
| *Diffusion Models:* | | | | | | | |
| LDM (Rombach et al., 2022) | 0.37 | 0.92 | 0.29 | 0.23 | 0.70 | 0.02 | 0.05 |
| SD1.5 (Rombach et al., 2022) | 0.43 | 0.97 | 0.38 | 0.35 | 0.76 | 0.04 | 0.06 |
| SD2.1 (Rombach et al., 2022) | 0.50 | 0.98 | 0.51 | 0.44 | 0.85 | 0.07 | 0.17 |
| SD-XL (Podell et al., 2023) | 0.55 | 0.98 | 0.74 | 0.39 | 0.85 | 0.15 | 0.23 |
| DALLE-2 (OpenAI, 2023) | 0.52 | 0.94 | 0.66 | 0.49 | 0.77 | 0.10 | 0.19 |
| DALLE-3 (Betker et al., 2023) | 0.67 | 0.96 | 0.87 | 0.47 | 0.83 | 0.43 | 0.45 |
| FLUX.1 Dev (Labs, 2024) | 0.66 | 0.98 | 0.81 | 0.74 | 0.79 | 0.22 | 0.45 |
| SD3.5-L (Esser et al., 2024) | 0.71 | 0.98 | 0.89 | 0.73 | 0.83 | 0.34 | 0.47 |
| SANA-1.5 4.8B (Xie et al., 2025) | 0.81 | 0.99 | 0.93 | 0.86 | 0.84 | 0.59 | 0.65 |
| SD3.5-M (Esser et al., 2024) | 0.63 | 0.98 | 0.78 | 0.50 | 0.81 | 0.24 | 0.52 |
| w/ Flow-GRPO (Liu et al., 2025) | 0.95 | **1.00** | 0.99 | 0.95 | 0.92 | **0.99** | 0.86 |
| w/ DGPO (Luo et al., 2025d) | **0.97** | **1.00** | **0.99** | **0.97** | **0.95** | **0.99** | **0.91** |
| *Few-Step Diffusion Models:* | | | | | | | |
| SD3.5-L-Turbo (4NFE) (Esser et al., 2024) | 0.70 | 0.94 | 0.84 | 0.55 | 0.79 | 0.58 | 0.56 |
| TDM-SD3.5-M (4NFE) (Luo et al., 2025c) | 0.61 | 0.99 | 0.77 | 0.49 | 0.79 | 0.23 | 0.44 |
| **SD3.5-M w/ TDM-R1 (Ours, 4NFE)** | **0.92** | **1.00** | **0.96** | **0.88** | **0.85** | **0.93** | **0.91** |

calculations as follows:

$$
\nabla_\theta L(\theta) = -\mathbb{E}_{k,t \geq t_k} \mathbb{E}_{p_\theta(\mathbf{x}_{t_k})} \mathbb{E}_{q(\mathbf{x}_t, \mathbf{x}_{t-1}|\mathbf{x}_{t_k})} \Bigg[
$$

$$
\beta \nabla_{\mathbf{x}_{t_k}} \log \frac{p_\phi(\mathbf{x}_{t-1}|\mathbf{x}_t)}{p_{\text{ref}}(\mathbf{x}_{t-1}|\mathbf{x}_t)} \tag{11}
$$

$$
+ \beta_g \lambda_t \left( s_{\text{fake}}(\mathbf{x}_t) - s_\psi(\mathbf{x}_t) \right) \Bigg] \frac{\partial \mathbf{x}_{t_k}}{\partial \theta}
$$

where we stop the gradient of $\phi$ to save the memory cost, and we found that it does not affect the performance. We defer the derivation to Appendix C. Following standard practice (Luo et al., 2023; 2025c; 2024a; Yin et al., 2023; Zhou et al., 2024), we adopt an online fake score $s_{fake}$ for estimating the score of student distribution, which is trained by denoising score matching.

**Remark.** 1) By joint training of the Few-Step Generator and Surrogate Reward, which establishes a synergistic loop: the Generator progressively produces higher-quality samples via maximize the Surrogate Reward, while the Surrogate Reward adapts to provide increasingly precise guidance by identifying favorable and unfavorable regions at each intermediate step per iteration. This Adaptive Surrogate Reward mechanism implements a GAN-like adversarial framework on self-generated samples, ultimately enabling effective reinforcement learning for few-step diffusion models with superior generation performance. 2) Our method applies to

$K$-step generators, meaning that **our method can also be used to reinforce one-step generators**, which corresponds to the case of $K = 1$.

In summary, TDM-R1 enables the utilization of large-scale online non-differentiable reward feedback, achieving effective reinforcement post-training for few-step generators. Starting from a pre-trained few-step generator based on Trajectory Distribution Matching (TDM), we alternately optimize the few-step generator to maximize the surrogate reward and minimize the reverse KL divergence, while the surrogate reward is trained through group preference optimization, and the fake score is optimized via the denoising matching objective. See Appendix E for pseudo algorithm.

## 4. Experiments

This section provides a comprehensive evaluation of TDM-R1. We primarily benchmark performance on two verifiable tasks with non-differentiable metrics: compositional image generation and visual text rendering (Tables 1 and 2). Qualitative comparisons are shown in Figure 2, and we demonstrate the method's effectiveness in aligning with human preferences (Figure 5). Additionally, we ablate key components of our approach (Figures 3 and 6).

### 4.1. Experimental Setup

**Evaluation Tasks.** We begin by pre-training the few-step TDM on SD3.5-M (Esser et al., 2024; Luo et al., 2025c), fol-

*Table 2.* **Performance on Compositional Image Generation and Visual Text Rendering** benchmarks. ImgRwd: ImageReward; UniRwd: UnifiedReward. We highlight the metric adopted for the training signal. [†] Results are taken from the original paper.

| Model | NFE | Verifiable Metric | | Image Quality | | Preference Score | | |
| --- | --- | --- | --- | --- | --- | --- | --- | --- |
| | | GenEval | OCR Acc. | Aesthetic | DeQA | ImgRwd | PickScore | UniRwd |
| SD3.5-M (Esser et al., 2024) | 80 | 0.63 | 0.59 | 5.39 | 4.07 | 0.87 | 22.34 | 3.33 |
| Flow-GRPO[†] (Liu et al., 2025) | 80 | 0.95 | — | 5.25 | 4.01 | 1.03 | 22.37 | 3.51 |
| DGPO[†] (Luo et al., 2025d) | 80 | 0.97 | — | 5.31 | 4.03 | 1.08 | 22.41 | 3.60 |
| Flow-GRPO[†] (Liu et al., 2025) | 80 | — | 0.92 | 5.32 | 4.06 | 0.95 | 22.44 | 3.42 |
| DGPO[†] (Luo et al., 2025d) | 80 | — | 0.96 | 5.37 | 4.09 | 1.02 | 22.52 | 3.48 |
| TDM-SD3.5-M (Luo et al., 2025c) | 4 | 0.61 | 0.55 | 5.41 | 4.05 | 0.99 | 22.36 | 3.30 |
| *Compositional Image Generation:* | | | | | | | | |
| $p_\phi$ in Surrogate Reward | 80 | 0.89 | 0.52 | 5.35 | 4.01 | 0.94 | 22.31 | 3.48 |
| **TDM-R1 (Ours)** | 4 | 0.92 | 0.59 | 5.42 | 4.07 | 1.11 | 22.39 | 3.55 |
| *Visual Text Rendering:* | | | | | | | | |
| $p_\phi$ in Surrogate Reward | 80 | 0.64 | 0.92 | 5.41 | 4.05 | 0.98 | 22.37 | 3.43 |
| **TDM-R1 (Ours)** | 4 | 0.67 | 0.95 | 5.45 | 4.11 | 1.09 | 22.62 | 3.51 |

*Table 3.* **Performance on reinforcing Z-Image**.

| Model | NFE | Verifiable Metric | | Model-based Metric | | | | |
| --- | --- | --- | --- | --- | --- | --- | --- | --- |
| | | GenEval | OCR Acc. | HPSv3 | Aesthetic | ImgRwd | PickScore | UniRwd |
| Z-Image | 100 | 0.66 | 0.74 | 7.32 | 5.35 | 0.62 | 19.98 | 3.64 |
| Z-Image-Turbo (DMDR) | 4 | 0.73 | 0.78 | 9.13 | 5.40 | 0.78 | 20.28 | 3.70 |
| **TDM-R1-Zimage (Ours)** | 4 | **0.77** | **0.79** | **9.90** | **5.49** | **0.94** | **20.45** | **3.75** |

lowed by evaluating our proposed TDM-R1's effectiveness in reinforcing the TDM across two valuable tasks with verifiable metrics: 1) *Compositional Image Generation*: assessed using GenEval (Ghosh et al., 2023), which encompasses six demanding compositional generation scenarios, including object counting, spatial relationships, and attribute binding; 2) *Visual Text Rendering*: measures the model's capability to synthesize text within generated images (Gong et al., 2025).

**Out-of-Domain Evaluation Metrics.** To ensure fair assessment and mitigate reward hacking—a phenomenon where models overfit to training reward signals at the expense of actual image quality—we utilize five image quality metrics excluded from training as out-of-domain evaluations: Aesthetic Score (Schuhmann et al., 2022), DeQA (You et al., 2025), ImageReward (Xu et al., 2023), PickScore (Kirstain et al., 2023), and UnifiedReward (Wang et al., 2025b). These metrics are computed on DrawBench (Saharia et al., 2022), a comprehensive benchmark comprising diverse prompts.

**4.2. Main Results**

**Quantitative Results.** Table 1 shows that our proposed TDM-R1 achieves competitive performance on GenEval, notably surpassing prior commercial closed-source SOTA GPT-4o and matches the performance of prior SOTA stan-

dard diffusion RL. *This promising performance is achieved while also showing improvement across various out-of-domain metrics (such as AeS, DeQA, PickScore, and Image Reward), as indicated by Table 2.* It is worth noting that, although previous SOTA of standard diffusion RL (e.g., Flow-GRPO and DGPO) achieved higher GenEval scores, they exhibited noticeable degradation in metrics measuring image quality. In contrast, our TDM-R1 even achieves higher image quality metrics compared to both the many-step base model and the few-step base model.

Beyond compositional image generation, Table 2 provides detailed evaluation results on visual text rendering, where DGPO similarly demonstrates significant improvements in both target optimization metrics and out-of-domain metrics. *Notably, we found that training on a verifiable metric (GenEval score or OCR score) can synergistically improve a completely different verifiable metric.* This is a surprising and encouraging signal, suggesting that we may be able to enhance the broad instruction-following capabilities of few-step diffusion models through a well-chosen proxy task.

**Qualitative Comparison.** We present qualitative results from our method and baselines, trained with GenEval's signal, in Figure 2. The visualizations demonstrate that TDM-R1 follows instructions more accurately than both

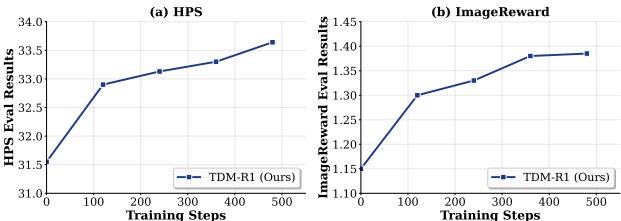

*Figure 5.* TDM-R1 Performance in human preference alignment.

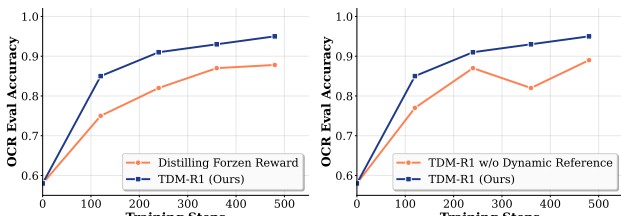

*Figure 6.* Comparison of visual text alignment across variants.

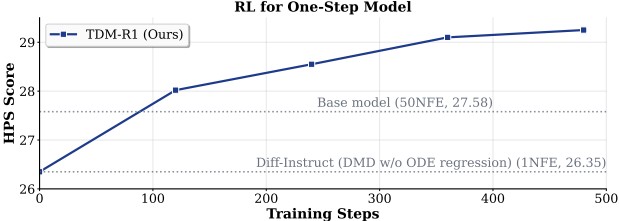

*Figure 7.* TDM-R1 in reinforcing one-step model.

the few-step baseline and the 80-NFE base model, while preserving high generation quality. See Appendix G for additional visual samples.

**Human Preference Alignment.** We also evaluate TDM-R1 on Human Preference Alignment. In this task, we employ ImageReward (Xu et al., 2023) and HPS (Wu et al., 2023) as the reward signal separately. Although these metrics are differentiable, we do not use their gradient. Both HPS and ImageReward are trained on large-scale human preference data, providing a comprehensive assessment of generation quality from a human-centric perspective. Figure 5 shows that, our TDM-R1 can effectively enhance few-step performance with these metrics that evaluate human preference.

**Human Preference Alignment on One-Step Model.** We evaluate TDM-R1 on aligning the one-step model to show its promising ability, using the diff-instruct as the base one-step model and HPSv2 as the reward signal. As shown in Figure 7, TDM-R1 effectively enhances one-step performance, notably surpassing the many-step base model.

**Alignment on Larger Model.** We further evaluate TDM-R1 on aligning a larger model, the powerful open-source Z-Image (Team et al., 2025) with 6B parameters, using HPSv3 as the reward signal. HPSv3 is a state-of-the-art reward model that serves as a robust metric for wide-spectrum image evaluation. Table 3 shows that TDM-R1 effectively enhances the performance of Z-Image across both in-domain and out-of-domain metrics, notably outperforming the many-

step Z-Image and the few-step Z-Image-Turbo.

**Comparison with the Surrogate Reward Model.** Since our Surrogate Reward is parameterized by a diffusion model $p_\phi$, an interesting experimental question arises: how does the performance of the diffusion model $p_\phi$ compare to our few-step generator TDM-R1? Table 2 shows that $p_\phi$ achieves substantially better task metrics than the base model. Notably, however, TDM-R1—which requires only 4 NFE for sampling—consistently outperforms $p_\phi$ (which requires 80 NFE) on both in-domain and out-of-domain metrics. This result may appear counterintuitive, given that TDM-R1 is trained using signals derived from the Surrogate Reward parameterized by $p_\phi$. Yet this observation aligns with findings from the LLM literature on DPO-like methods: improving a model's performance as a reward signal does not necessarily translate to improved generation performance. The key insight is that effective training of TDM-R1 depends not on $p_\phi$'s generation capabilities, but rather on the quality of the reward signal it provides. These results demonstrate that our Surrogate Reward has been effectively tailored as a per-step reward model, enabling superior reinforcement fine-tuning performance for TDM-R1.

### 4.3. Ablation Study

**TDM-R1 v.s. Direct RL Loss Combination.** A fairly straightforward and natural baseline is to directly combine the distillation loss with an RL loss designed for standard diffusion models to reinforce the few-step generator. We adopt DGPO (Luo et al., 2025d), a recent state-of-the-art RL method for diffusion models, as the accompanying RL loss and combine it with TDM's distillation loss as a potential baseline, termed TDM w/ direct RL loss. As shown in Figures 3 and 4, while this baseline achieves modest performance improvements in the early stages (though still far inferior to TDM-R1), it produces blurry images and suffers from performance degradation in later training stages. This decline in image quality stems from a fundamental mismatch: the denoising objective inherent in standard diffusion RL is incompatible with reverse KL divergence minimization in distillation, as the direct denoising behavior enforced by RL loss may be suboptimal for few-step generation.

**Dynamic Surrogate Reward v.s. Frozen Well-Trained Reward.** Our proposed TDM-R1 jointly trains the Dynamic Surrogate Reward and the few-step generator throughout the training process. We compared this approach against a notable baseline: directly using the reward model trained by DGPO. This baseline can be viewed as using an RL-trained diffusion model as a teacher to distill a few-step student. As shown in Figure 6, the Dynamic Surrogate Reward achieves faster reward growth and superior final performance compared to distillation with a frozen reward. We attribute this improvement to two factors. First, the Dynamic Surrogate Reward can dynamically identify regions where the few-step

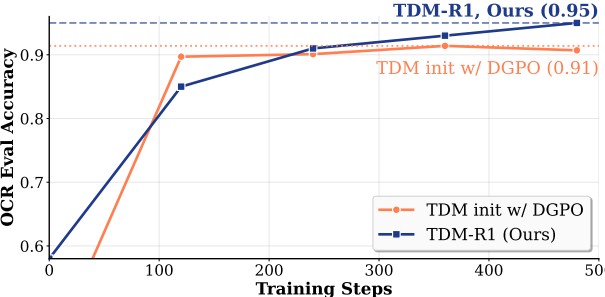

*Figure 8.* Compare TDM-R1 with direct distillation.

student performs well or poorly, enabling more targeted optimization—a capability that the frozen reward lacks. Second, a distribution gap exists between the few-step generator's outputs and the data distribution on which the frozen reward was originally trained, which can lead to suboptimal guidance and degraded performance.

**Comparison with Distilled RL Diffusion.** We also compare TDM-R1 against a competitive baseline: initializing with an RL-finetuned diffusion model (i.e., DGPO) followed by distillation via TDM. As shown in Figure 8, TDM initialized with DGPO converges quickly in the early stages, but its performance ceiling is constrained by the teacher model, causing it to plateau rapidly. In contrast, TDM-R1 continuously incorporates reward signals throughout training, ultimately achieving superior performance. Although this baseline is beyond our scope — our goal is to directly post-train few-step diffusion models rather than first training a teacher and then distilling a few-step student, which introduces additional pipeline complexity — this ablation clearly demonstrates that *TDM-R1 is not only more elegant as a paradigm, directly reinforcing the well-trained few-step student model, but also achieves stronger final performance.*

**Effect of the Deterministic Path.** A core feature of our method is its ability to leverage deterministic trajectories for step-by-step reward feedback. While stochastic trajectories can theoretically achieve the same goal, they introduce greater variance in the feedback signal. To evaluate this trade-off, we compared our approach against a variant using stochastic sampling, termed TDM-R1 w/ stochastic sampling. As shown in Figure 3, deterministic trajectories yield faster convergence and superior performance.

**Effect of the Dynamic Reference Model.** We propose using an EMA of $p_\phi$ as the reference model during surrogate reward learning. This dynamic reference model enables effective optimization. Our experiments confirm the effectiveness of this design: as shown in Figure 6, replacing the dynamic reference model with a static one results in degraded performance and reduced training stability.

## 5. Related Works

Recent efforts to improve diffusion models through reinforcement learning have explored three main strategies. The first fine-tunes diffusion on high-quality image-prompt pairs (Dai et al., 2023; Podell et al., 2023). The second optimizes explicit rewards, either by evaluating the final outputs of multi-step synthesis (Prabhudesai et al., 2023; Clark et al., 2023; Lee et al., 2023; Ho et al., 2022; Luo et al., 2025a;b) or through policy gradient methods (Fan et al., 2024; Black et al., 2023; Ye et al., 2024). The third bypasses explicit rewards entirely, instead learning directly from preference data, as in Diffusion-DPO (Wallace et al., 2024) and Diffusion-KTO (Yang et al., 2024). More recently, GRPO has been successfully adapted to diffusions (Liu et al., 2025; Xue et al., 2025b; Luo et al., 2025d), showing strong scalability and notable performance gains. Nevertheless, all these approaches target standard diffusions. Our work tackles a distinct problem: reinforcing the few-step diffusion models.

As for few-step generative models, recent work (Luo, 2024; Luo et al., 2024b; Li et al., 2024; Ren et al., 2024; Luo et al., 2025a;b) has introduced methods for training preference-aligned few-step text-to-image models, achieving promising results. Nevertheless, these existing approaches for few-step models are restricted to differentiable reward functions, with narrow applications in cases of non-differentiable rewards. Some others theoretically support non-differentiable rewards (Miao et al., 2025; Jiang et al., 2025) by combining standard RL methods, but they have not validated their effectiveness at scales. Furthermore, applying standard diffusion RL methods to few-step models would lead to blurry generation and suboptimal performance, as discussed in Section 3.2 and illustrated in Figures 3 and 4. To our best knowledge, our TDM-R1 is the first work tackling RL with non-differentiable rewards for few-step text-to-image generative models at scales.

We defer additional discussion on related works in Appendix D.

## 6. Conclusion

In this paper, we introduced TDM-R1, a novel reinforcement learning paradigm that enables few-step diffusion models to effectively leverage non-differentiable reward feedback. By building upon Trajectory Distribution Matching and decoupling the learning process into surrogate reward learning and generator optimization, TDM-R1 addresses a fundamental limitation of existing approaches that rely exclusively on differentiable reward signals. Our extensive experiments demonstrate that TDM-R1 consistently achieves state-of-the-art reinforcement learning performance for few-step text-to-image models. Most notably, TDM-R1 enables 4-step models to surpass their expensive 80-NFE base counterparts, with improvements on GenEval from 61% to 92%—exceeding both the 80-NFE base model at 63%. These results validate that few-step diffusion models can successfully incorporate large-scale online non-differentiable reward signals without requiring additional ground-truth image data.

## Acknowledgment

This work is partially supported by National Key R&D Program of China under Grant No. 2024YFA1012700, by the National Natural Science Foundation of China (NSFC) under Grant No. 62402410, by Guangdong Provincial Project (No. 2023QN10X025), and by HKUST(GZ) Kunpeng&Ascend Center of Cultivation.

## Impact Statement

This paper presents work whose goal is to advance the field of Machine Learning. There are many potential societal consequences of our work, none which we feel must be specifically highlighted here.

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

## A. Noation Summary

*Table 4.* Notation summary for TDM-R1.

| Symbol | Description |
|---|---|
| $\theta$ | Parameters of the $K$-step generator. |
| $\psi$ | Parameters of the frozen pretrained diffusion model. |
| $\phi$ | Parameters of the diffusion-parameterized surrogate reward. |
| $p_{\theta,k}(\mathbf{x}_t)$ | Forward-perturbed student marginal, $\int q(\mathbf{x}_t|\mathbf{x}_{t_k})\, p_\theta(\mathbf{x}_{t_k})\, d\mathbf{x}_{t_k}$. |
| $p_\psi$ | Frozen Pretrained teacher diffusion model. |
| $p_\phi$ | Diffusion model that parameterizes the surrogate reward. |
| $p_{\text{ref}}$ | Reference distribution in the the surrogate reward. |
| $q(\mathbf{x}_t|\mathbf{x}_0)$ | Forward diffusion kernel, $\mathcal{N}(\alpha_t\mathbf{x}_0, \sigma_t^2\mathbf{I})$. |
| $q(\mathbf{x}_{t-1}|\mathbf{x}_t, \mathbf{x}_{t_k})$ | DDIM-style posterior used in $\Delta\mathrm{KL}_\phi$. |

## B. Limitation and Future Work

While TDM-R1 achieves strong empirical results across diverse rewards and backbones, several theoretical aspects remain open for future investigation. In particular, our surrogate reward objective inherits a Jensen's inequality relaxation that is common to the broader family of Diffusion-DPO–style methods, and a unified theoretical understanding of this relaxation across such methods would be a valuable direction. In addition, the interesting phenomenon that our 4-step student surpasses its 80-step teacher $p_\phi$ is currently supported by intuitive explanations, and we believe a rigorous theoretical characterization would further illuminate the mechanism behind this result. We view these as exciting opportunities for follow-up work, and hope that our empirical findings can serve as a useful foundation for deeper theoretical understanding of RL for few-step generative models.

## C. Derivation

### C.1. Derivation of Equation (9)

For simplicity and without loss of generality, we omit the conditioning on $\mathbf{c}$ throughout this derivation. By substituting the surrogate reward parameterization from Equation (7) into the Bradley-Terry objective in Equation (8), we obtain the initial training objective:

$$
\begin{aligned}
L(\phi) = -\mathbb{E}_k \mathbb{E}_{(\mathcal{G}_k^+, \mathcal{G}_k^-)\sim\mathcal{D}} \log \sigma \Bigg( & \sum_{\mathbf{x}_{t_k}\in\mathcal{G}_k^+} \mathbb{E}_{q(\mathbf{x}_{t_k+1:T}|\mathbf{x}_{t_k})}\beta w(\mathbf{x}_{t_k}) \left[\log \frac{p_\phi(\mathbf{x}_{t_k:T})}{p_{\text{ref}}(\mathbf{x}_{t_k:T})} + \log Z\right] \\
& - \sum_{\mathbf{x}_{t_k}\in\mathcal{G}_k^-} \mathbb{E}_{q(\mathbf{x}_{t_k+1:T}|\mathbf{x}_{t_k})}\beta w(\mathbf{x}_{t_k}) \left[\log \frac{p_\phi(\mathbf{x}_{t_k:T})}{p_{\text{ref}}(\mathbf{x}_{t_k:T})} + \log Z\right] \Bigg).
\end{aligned}
\tag{12}
$$

The partition function terms cancel because the weights sum to zero over the full group. Specifically, since the weights are defined as absolute normalized advantages and the partition into positive and negative groups is based on the sign of the advantage, we have $\sum_{\mathbf{x}_{t_k}\in\mathcal{G}_k^+} w(\mathbf{x}_{t_k}) - \sum_{\mathbf{x}_{t_k}\in\mathcal{G}_k^-} w(\mathbf{x}_{t_k}) = \sum_{\mathbf{x}_{t_k}\in\mathcal{G}_k} A(\mathbf{x}_{t_k}) = \sum_{\mathbf{x}_{t_k}\in\mathcal{G}_k} \frac{r(\mathbf{x}_k)-r_{mean}}{r_{std}} = 0$, where

$r_{mean} = \frac{\sum_{\mathbf{x}_{t_k} \in \mathcal{G}_k} r(\mathbf{x}_k)}{|G|}$. This simplifies the objective to:

$$L(\phi) = -\mathbb{E}_{(\mathcal{G}^+,\mathcal{G}^-)\sim\mathcal{D}} \log \sigma\left(\beta\left[\sum_{\mathbf{x}_{t_k}\in\mathcal{G}^+} w(\mathbf{x}_{t_k})\mathbb{E}_{q(\mathbf{x}_{t_k+1:T}|\mathbf{x}_{t_k})} \log \frac{p_\phi(\mathbf{x}_{t_k:T})}{p_{\text{ref}}(\mathbf{x}_{t_k:T})}\right.\right.$$
$$\left.\left. - \sum_{\mathbf{x}_{t_k}\in\mathcal{G}^-} w(\mathbf{x}_{t_k})\mathbb{E}_{q(\mathbf{x}_{t_k+1:T}|\mathbf{x}_{t_k})} \log \frac{p_\phi(\mathbf{x}_{t_k:T})}{p_{\text{ref}}(\mathbf{x}_{t_k:T})}\right]\right). \tag{13}$$

Next, we decompose the trajectory-level log-ratio using the Markov property of the diffusion process. For any trajectory $\mathbf{x}_{t_k:T}$, the joint distribution factorizes as a product of transition probabilities:

$$\log \frac{p_\phi(\mathbf{x}_{t_k:T})}{p_{\text{ref}}(\mathbf{x}_{t_k:T})} = \sum_{t=t_k+1}^{T} \log \frac{p_\phi(\mathbf{x}_{t-1}|\mathbf{x}_t)}{p_{\text{ref}}(\mathbf{x}_{t-1}|\mathbf{x}_t)}. \tag{14}$$

Substituting this factorization and rewriting the sum over timesteps as an expectation, we obtain:

$$L(\phi) = -\mathbb{E}_{(\mathcal{G}^+,\mathcal{G}^-)\sim\mathcal{D}} \log \sigma\left(\beta(T-t_k)\mathbb{E}_{t\sim\mathcal{U}[t_k+1,T]}\mathbb{E}_{q(\mathbf{x}_t|\mathbf{x}_{t_k}),q(\mathbf{x}_{t-1}|\mathbf{x}_t,\mathbf{x}_{t_k})}\right.$$
$$\left. \times \left[\sum_{\mathbf{x}_{t_k}\in\mathcal{G}^+} w(\mathbf{x}_{t_k}) \log \frac{p_\phi(\mathbf{x}_{t-1}|\mathbf{x}_t)}{p_{\text{ref}}(\mathbf{x}_{t-1}|\mathbf{x}_t)} - \sum_{\mathbf{x}_{t_k}\in\mathcal{G}^-} w(\mathbf{x}_{t_k}) \log \frac{p_\phi(\mathbf{x}_{t-1}|\mathbf{x}_t)}{p_{\text{ref}}(\mathbf{x}_{t-1}|\mathbf{x}_t)}\right]\right). \tag{15}$$

To obtain a tractable upper bound, we apply Jensen's inequality. Since $-\log\sigma(\cdot)$ is a convex function, moving the expectation over $t$ and $q(\mathbf{x}_t|\mathbf{x}_{t_k})$ inside the sigmoid yields:

$$L(\phi) \leq \mathbb{E}_{(\mathcal{G}^+,\mathcal{G}^-)\sim\mathcal{D}}\mathbb{E}_{t,q(\mathbf{x}_t|\mathbf{x}_{t_k})}\left[-\log\sigma\left(\beta(T-t_k)\mathbb{E}_{q(\mathbf{x}_{t-1}|\mathbf{x}_t,\mathbf{x}_{t_k})}\right.\right.$$
$$\left.\left. \times \left[\sum_{\mathbf{x}_{t_k}\in\mathcal{G}^+} w(\mathbf{x}_{t_k}) \log \frac{p_\phi(\mathbf{x}_{t-1}|\mathbf{x}_t)}{p_{\text{ref}}(\mathbf{x}_{t-1}|\mathbf{x}_t)} - \sum_{\mathbf{x}_{t_k}\in\mathcal{G}^-} w(\mathbf{x}_{t_k}) \log \frac{p_\phi(\mathbf{x}_{t-1}|\mathbf{x}_t)}{p_{\text{ref}}(\mathbf{x}_{t-1}|\mathbf{x}_t)}\right]\right)\right]. \tag{16}$$

Finally, we rewrite the log-ratios in terms of KL divergences. For any distribution $q$ and models $p_\phi$, $p_{\text{ref}}$, we have the following identity:

$$\mathbb{E}_{q(\mathbf{x}_{t-1}|\mathbf{x}_t,\mathbf{x}_{t_k})} \log \frac{p_\phi(\mathbf{x}_{t-1}|\mathbf{x}_t)}{p_{\text{ref}}(\mathbf{x}_{t-1}|\mathbf{x}_t)} = \text{KL}\big(q(\mathbf{x}_{t-1}|\mathbf{x}_t,\mathbf{x}_{t_k})\|p_{\text{ref}}(\mathbf{x}_{t-1}|\mathbf{x}_t)\big) - \text{KL}\big(q(\mathbf{x}_{t-1}|\mathbf{x}_t,\mathbf{x}_{t_k})\|p_\phi(\mathbf{x}_{t-1}|\mathbf{x}_t)\big). \tag{17}$$

Substituting this identity and rearranging terms, we arrive at the final upper bound:

$$L(\phi) \leq \mathbb{E}_{(\mathcal{G}_k^+,\mathcal{G}_k^-)\sim\mathcal{D}_k,k}\mathbb{E}_{t,q(\mathbf{x}_t|\mathbf{x}_{t_k})} \log \sigma\left(-\beta(T-t_k)\left\{\right.\right.$$
$$\sum_{\mathbf{x}_0\in\mathcal{G}_k^+} w(\mathbf{x}_{t_k})\Big[\text{KL}\big(q(\mathbf{x}_{t-1}|\mathbf{x}_t,\mathbf{x}_{t_k})\|p_\phi(\mathbf{x}_{t-1}|\mathbf{x}_t)\big) - \text{KL}\big(q(\mathbf{x}_{t-1}|\mathbf{x}_t,\mathbf{x}_{t_k})\|p_{\text{ref}}(\mathbf{x}_{t-1}|\mathbf{x}_t)\big)\Big]$$
$$\left.\left. - \sum_{\mathbf{x}_{t_k}\in\mathcal{G}_k^-} w(\mathbf{x}_{t_k})\Big[\text{KL}\big(q(\mathbf{x}_{t-1}|\mathbf{x}_t,\mathbf{x}_{t_k})\|p_\phi(\mathbf{x}_{t-1}|\mathbf{x}_t)\big) - \text{KL}\big(q(\mathbf{x}_{t-1}|\mathbf{x}_t,\mathbf{x}_{t_k})\|p_{\text{ref}}(\mathbf{x}_{t-1}|\mathbf{x}_t)\big)\Big]\right\}\right). \tag{18}$$

This completes the derivation of Equation (9).

### C.2. Derivation of Equation (11)

We compute the gradient of the objective in Equation (10) by separately deriving the gradients of the reward maximization and KL regularization terms.

**Step 1: Reformulating the Surrogate Reward.** We first re-write the surrogate reward defined in Equation (7) as follows:

$$
\begin{aligned}
\tilde{r}_\phi(\mathbf{x}_{t_k}, \mathbf{c}) &= \beta \mathbb{E}_{q(\mathbf{x}_{t_{k+1}:T}|\mathbf{x}_{t_k})} \log \frac{p_\phi(\mathbf{x}_{t_k:T}|\mathbf{c})}{p_{\mathrm{ref}}(\mathbf{x}_{t_k:T}|\mathbf{c})} + \beta \log Z \\
&= \beta \mathbb{E}_{q(\mathbf{x}_{t_{k+1}:T}|\mathbf{x}_{t_k})} \sum_{t=t_k+1}^{T} \log \frac{p_\phi(\mathbf{x}_{t-1}|\mathbf{x}_t)}{p_{\mathrm{ref}}(\mathbf{x}_{t-1}|\mathbf{x}_t)} + \beta \log Z.
\end{aligned}
\tag{19}
$$

By converting the summation over timesteps into an expectation with respect to a uniform distribution over $t \geq t_k$, we obtain:

$$
\begin{aligned}
\tilde{r}_\phi(\mathbf{x}_{t_k}, \mathbf{c}) &= \beta(T - t_k) \mathbb{E}_{t \geq t_k} \mathbb{E}_{q(\mathbf{x}_{t_{k+1}:T}|\mathbf{x}_{t_k})} \log \frac{p_\phi(\mathbf{x}_{t-1}|\mathbf{x}_t)}{p_{\mathrm{ref}}(\mathbf{x}_{t-1}|\mathbf{x}_t)} + \beta \log Z \\
&= \beta(T - t_k) \mathbb{E}_{t \geq t_k} \mathbb{E}_{q(\mathbf{x}_t|\mathbf{x}_{t_k})} \mathbb{E}_{q(\mathbf{x}_{t-1}|\mathbf{x}_t, \mathbf{x}_{t_k})} \log \frac{p_\phi(\mathbf{x}_{t-1}|\mathbf{x}_t)}{p_{\mathrm{ref}}(\mathbf{x}_{t-1}|\mathbf{x}_t)} + \beta \log Z,
\end{aligned}
\tag{20}
$$

where the last equality follows from marginalizing the joint distribution $q(\mathbf{x}_{t_{k+1}:T}|\mathbf{x}_{t_k})$ to the relevant variables $\mathbf{x}_t$ and $\mathbf{x}_{t-1}$.

**Step 2: Gradient of the Reward Term.** Applying the chain rule through the reparameterized sampling process, the gradient of the reward term with respect to $\theta$ is given by:

$$
\nabla_\theta \mathbb{E}_{k, p_\theta(\mathbf{x}_{t_k})} \tilde{r}_\phi(\mathbf{x}_{t_k}, \mathbf{c}) = \beta(T - t_k) \mathbb{E}_{k, t \geq t_k} \mathbb{E}_{p_\theta(\mathbf{x}_{t_k})} \mathbb{E}_{q(\mathbf{x}_t, \mathbf{x}_{t-1}|\mathbf{x}_{t_k})} \left[ \nabla_{\mathbf{x}_{t_k}} \log \frac{p_\phi(\mathbf{x}_{t-1}|\mathbf{x}_t)}{p_{\mathrm{ref}}(\mathbf{x}_{t-1}|\mathbf{x}_t)} \right] \frac{\partial \mathbf{x}_{t_k}}{\partial \theta}.
\tag{21}
$$

Note that the gradient with respect to $\log Z$ vanishes since the partition function is independent of $\mathbf{x}_t$.

**Step 3: Gradient of the KL Regularization Term.** The gradient of the marginal-level KL divergence term follows from Equation (2).

**Step 4: Combining the Gradients.** Combining the results from Steps 2 and 3, and incorporating the negative sign from the reward maximization term in Equation (10), we arrive at the final gradient expression:

$$
\begin{aligned}
\nabla_\theta L(\theta) = -\mathbb{E}_{k, t \geq t_k} \mathbb{E}_{p_\theta(\mathbf{x}_{t_k})} \mathbb{E}_{q(\mathbf{x}_t, \mathbf{x}_{t-1}|\mathbf{x}_{t_k})} \Bigg[ \\
\beta(T - t_k) \nabla_{\mathbf{x}_{t_k}} \log \frac{p_\phi(\mathbf{x}_{t-1}|\mathbf{x}_t)}{p_{\mathrm{ref}}(\mathbf{x}_{t-1}|\mathbf{x}_t)} \\
+ \beta_g \lambda_t \left( s_{\mathrm{fake}}(\mathbf{x}_t) - s_\psi(\mathbf{x}_t) \right) \Bigg] \frac{\partial \mathbf{x}_{t_k}}{\partial \theta},
\end{aligned}
\tag{22}
$$

which completes the derivation of Equation (11).

## D. Additional Related Works

**Few-Step Text-to-Image Diffusion Sampling** Although training-free methods for accelerating diffusion model sampling have progressed substantially (Lu et al., 2022; Zhao et al., 2023; Xue et al., 2024; Si et al., 2024; Ma et al., 2024; Li et al., 2025), diffusion distillation (Luo, 2023) remains essential for high-quality few-step generation. In general, distilled models perform one or a few more transformations that map random noise to images. The most widely adopted distillation strategies for few-step diffusion sampling fall into two categories: trajectory matching (Song et al., 2023; Kim et al., 2023; Song & Dhariwal; Geng et al., 2024; Salimans et al., 2024) and distribution matching (Luo et al., 2023; Yin et al., 2023; Zhou et al., 2024; Luo et al., 2024a; Sauer et al., 2023; Xu et al., 2024; Luo et al., 2024c; Wang et al., 2025a). More recently, trajectory distribution matching (TDM) (Luo et al., 2025c) has demonstrated strong few-step performance.

---

**Algorithm 1** TDM-R1

---

**Require:** Few-step diffusion model $p_\theta$, surrogate reward model $\tilde{r}_\phi$, reward function $r$, group size $G$, number of iterations $N$, number of denoising steps $K$.

**Ensure:** Reinforced few-step model $p_\theta$.

1: **for** $n \leftarrow 1$ **to** $N$ **do**
2:     *// Sample conditioning and generate group*
3:     Sample conditioning $\mathbf{c} \sim \mathcal{D}_c$
4:     Generate group $\mathcal{G} = \{\mathbf{x}_0^{(1)}, \dots, \mathbf{x}_0^{(G)}\}$ by sampling from $p_{\theta-}(\cdot \mid \mathbf{c})$ with $K$ steps
5:     *// Access reward feedback*
6:     Compute rewards: $r_i \leftarrow r(\mathbf{c}, \mathbf{x}_0^{(i)})$ for $i = 1, \dots, G$
7:     *// Update models*
8:     Update surrogate reward $\tilde{r}_\phi$ via Eq. equation 9
9:     Update fake score $s_{\text{fake}}$ via denoising on perturbed generated samples (Equation (1)).
10:     Update few-step generator $p_\theta$ via Eq. equation 10
11: **end for**

---

# E. Algorithm

# F. Experiment details

**Visual Text Rendering.** Adopting the evaluation protocol from TextDiffuser (Chen et al., 2023) and the experimental framework of Flow-GRPO, we assess how accurately models can render textual content within generated images. Prompts follow a standardized template: "A sign that says 'text'", where 'text' denotes the target string to be visually rendered. Text fidelity (Gong et al., 2025) is computed as:

$$r = \max(1 - N_e/N_{\text{ref}}, 0)$$

Here, $N_e$ represents the minimum edit distance between the rendered output and the target text, while $N_{\text{ref}}$ indicates the character length of the quoted string in the prompt.

**Setup Details.** During training, we generate 24 samples per group. We employ Flow-DPM-Solver (Xie et al., 2025) with 4 sampling steps for both training rollouts and inference. We apply LoRA fine-tuning with rank 32, and set $\beta = 100$ as the default. Our default choice of $\beta_g$ is set such that the ratio of the gradient of the reward term to the gradient of the KL regularization term is $2 : 1$. All experiments are conducted at 512 resolution, with GPU hours reported in A100 equivalents.

**Details of the out-of-domain evaluation metrics.** We describe the specific out-of-domain metrics employed for quality assessment. The `aesthetic score` (Schuhmann et al., 2022) uses a CLIP-based linear regression model to quantify the visual appeal of generated images. To evaluate image quality degradation, we employ the `DeQA score` (You et al., 2025), which utilizes a multimodal large language model architecture to measure how various imperfections—such as distortions, textural degradation, and low-level visual artifacts—affect overall perceived image quality. `ImageReward` (Kirstain et al., 2023) assesses both visual quality and text-image correspondence, offering a holistic evaluation of generation quality from a human-centered standpoint. `ImageReward` (Xu et al., 2023) functions as a comprehensive human preference model for text-to-image generation, evaluating multiple aspects including text-visual coherence and generation fidelity. Lastly, `UnifiedReward` (Wang et al., 2025b) represents a state-of-the-art advancement in this domain. This unified reward framework is capable of evaluating both multimodal understanding and generation tasks, and has achieved superior performance on human preference assessment benchmarks compared to prior methods.

# G. Additional Qualitative Comparison

We present additional visual comparison in Figure 9.

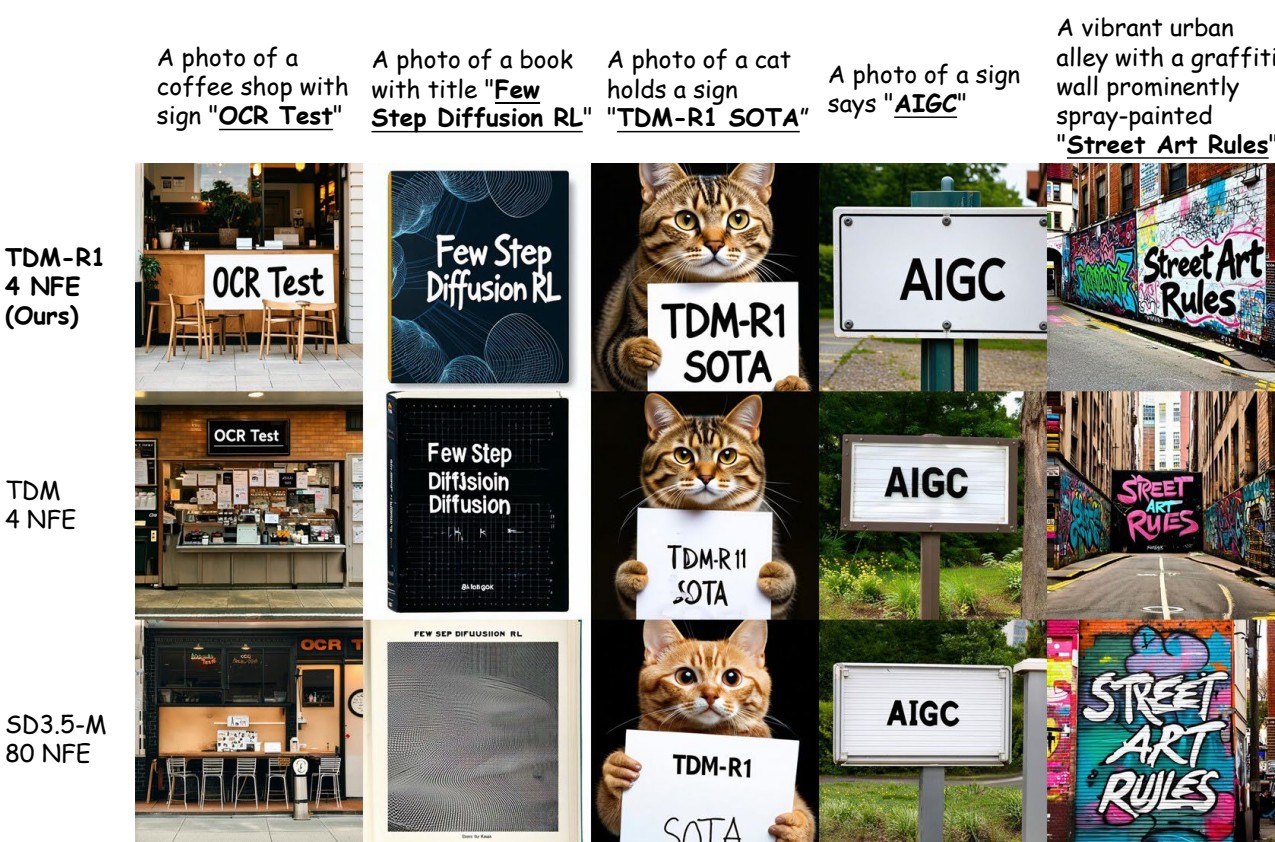

*Figure 9.* Qualitative comparisons of TDM-R1 against competing baselines. The same initial noise is used to generate all images.

