# OpenReview forum: "TDM-R1: Reinforcing Few-Step Diffusion Models with Non-Differentiable Reward"
_ICML.cc/2026/Conference — ICML 2026 regular_

### Official Review · Reviewer_axpY · 2026-03-02

**Soundness:** 2
**Presentation:** 3
**Significance:** 2
**Originality:** 2
**Overall Recommendation:** 4
**Confidence:** 2

**Summary:**

The provided manuscript introduces **TDM-R1**, a novel reinforcement learning (RL) paradigm designed to optimize few-step diffusion models using non-differentiable reward signals.

**Context and Problem:**
While few-step generative models offer rapid inference, existing RL techniques designed to improve them rely heavily on differentiable reward models. This strict requirement excludes many practical, real-world reward signals that are inherently non-differentiable, such as human binary preferences, OCR accuracy for text rendering, and discrete object counts. Additionally, applying standard multi-step diffusion RL methods directly to few-step models is fundamentally incompatible and often results in degraded, blurry image quality.

**Proposed Methodology:**
To resolve these challenges, the authors build upon a few-step model called Trajectory Distribution Matching (TDM). TDM-R1 decouples the RL process into two primary mechanisms:

- **Surrogate Reward Learning:** By leveraging the deterministic sampling trajectories inherent to TDM, the method is able to accurately assign unbiased reward estimates to intermediate denoising steps. It trains a diffusion-parameterized "Surrogate Reward" via group-based preference optimization, providing fine-grained, step-by-step guidance. It also utilizes a dynamic Exponential Moving Average (EMA) reference model to stabilize training.

- **Few-Step Generator Optimization:** The few-step generator is trained to maximize this surrogate reward while simultaneously applying a marginal-level reverse KL regularization. This regularization ensures the generated samples remain grounded in the high-quality base distribution, preventing image degradation.

**Key Contributions and Results:**

- **Methodological Innovation:** TDM-R1 successfully establishes a unified post-training RL framework for few-step text-to-image models that handles free-form, generic rewards without requiring ground-truth image data.

- **Benchmark Success:** The method dramatically boosts the performance of a 4-step model (TDM-SD3.5-M) on the rigorous GenEval benchmark from 61% to 92%. This performance significantly outperforms its expensive 80-step base model (63%) and matches or exceeds commercial state-of-the-art models like GPT-4o (84%).

- **Preservation of Image Quality:** TDM-R1 achieves these strict instruction-following improvements (like visual text rendering and compositional alignment) while maintaining or even enhancing out-of-domain visual quality metrics (e.g., Aesthetic Score, ImageReward), successfully avoiding reward hacking.

**Compliance With Llm Reviewing Policy:**

Affirmed.

**Final Justification:**

**Final Recommendation:** Weak Accept

**Justification:**
I have thoroughly evaluated the authors' rebuttal and have decided to raise my final recommendation. The rebuttal effectively addressed my primary methodological concerns, convincing me that the practical and empirical contributions of this work outweigh its theoretical limitations.

**Originality & Significance (Strengths):** The manuscript tackles a highly significant bottleneck in generative AI: the integration of non-differentiable rewards into few-step diffusion models. The proposed mechanism (TDM-R1) is highly original in its use of deterministic trajectories to bypass variance explosion. The empirical leap demonstrated on GenEval (from 61% to 92%)—without sacrificing out-of-domain visual fidelity—is a substantial and practically useful contribution to the field.

**Soundness, Clarity & Rebuttal Impact (Weaknesses Addressed):** My initial review raised critical concerns regarding the validity of the comparative baseline (DGPO) and the theoretical justification for the model's performance paradox (the 4-step student outperforming its 80-step teacher).

* **The Baseline Concern (Resolved):** The authors successfully defended their choice of the DGPO baseline. By clarifying that there are currently no existing non-differentiable RL frameworks specifically designed for few-step models, they proved that this baseline was not a "strawman," but a necessary and fair reflection of the current literature's limits. This significantly strengthens the validity of their empirical claims.
* **Theoretical Limitations (Acknowledged):** Regarding the performance paradox and the unquantified Jensen bound, the authors honestly acknowledged the lack of a rigorous mathematical proof. They offered a highly plausible intuitive explanation (the 4-step model avoids the sequential error accumulation of the 80-step teacher) and cited established community precedents. While the theoretical chassis remains somewhat fragile, I accept this as a valuable empirical finding rather than a fatal flaw.
* **Scope & Compute (Resolved):** The new data provided in the rebuttal regarding compatibility with stochastic trajectories (DMD2) and the negligible memory overhead (68G vs 64.5G) adequately resolved my concerns about the framework's generalizability and cost.

**Conclusion:**
The rebuttal was highly effective. It changed my evaluation by validating the experimental design and clarifying the boundaries of the theoretical claims. The work is technically solid and presents a high-impact solution for the research community. I am raising my score, with the strong expectation that the authors explicitly incorporate the theoretical limitations and nuances discussed in the rebuttal into the final camera-ready manuscript.

**Key Questions For Authors:**

**1. The Unquantified Mathematical Bound in Equation 8** In Appendix A.1, you employ Jensen's inequality to construct a tractable upper bound for the surrogate reward learning objective. However, the manuscript provides no empirical quantification of the tightness of this bound during training. How significant is the divergence between the true log-ratio and this approximated bound across the 500 training steps?

- **Impact on Evaluation:** If you can provide empirical evidence (e.g., tracking the KL gap) demonstrating that this bound remains tight throughout optimization, it would validate the mathematical soundness of the Surrogate Reward. This would remove a major theoretical vulnerability and substantially elevate the paper's soundness score. Failure to quantify this gap leaves the objective structurally compromised.

**2. The Surrogate Performance Paradox** Table 2 demonstrates that the 4-step generator (TDM-R1) consistently outperforms the 80-step model ($p_\phi$) that parameterizes its very own training signal across both in-domain and out-of-domain metrics. Your justification relies heavily on a conceptual analogy to the LLM DPO literature. Can you provide a mathematically rigorous explanation, or a targeted ablation study isolating the gradient flow, to explain exactly *how* the generator filters out the base model's inherent generative limitations in a continuous diffusion space?

- **Impact on Evaluation:** Relying on superficial cross-domain analogies is academically insufficient. Providing a mechanical or mathematical proof for this phenomenon would resolve the largest analytical gap in the paper. This would shift my view from seeing this as an "empirical anomaly" to a "theoretically sound contribution," heavily favoring acceptance.

**3. The Validity of the DGPO Baseline** Your primary ablation benchmarks TDM-R1 against a "direct RL loss combination" using DGPO. However, you explicitly concede that the denoising objectives inherent to standard diffusion RL are incompatible with few-step generation, inevitably causing blurry outputs. Why was the proposed method not benchmarked against contemporary frameworks explicitly designed for few-step alignment (such as those you cite in Section 5, e.g., Diff-Instruct\* or Reward-Instruct ), rather than a structurally hobbled baseline?

- **Impact on Evaluation:** Defeating a baseline that is mathematically guaranteed to fail does not validate your architecture. If you can present an empirical comparison against a genuinely optimized, competitive few-step alignment baseline and demonstrate that TDM-R1 still dominates, the empirical claims will become unassailable. This would directly address the "strawman" critique and justify a stronger recommendation.

**Limitations:**

1. **Architectural Dependency**: Acknowledge that the framework relies heavily on the deterministic trajectories of Trajectory Distribution Matching (TDM) and currently lacks proven generalizability to other dominant few-step distillation architectures (like Consistency Models) .
2. **Theoretical Vulnerability**: Transparently state the limitation of the unquantified upper bound introduced via Jensen's inequality in your surrogate reward derivation (Equation 15) .
3. **Computational Overhead**: Address the computational cost and memory footprint required to jointly train a dynamic 80-NFE surrogate reward (via an Exponential Moving Average) alongside the few-step generator during the reinforcement phase .

**Strengths And Weaknesses:**

### Strengths:
**1. High-Value Problem Identification & Scope:** The manuscript correctly identifies a critical bottleneck in contemporary generative alignment: the strict reliance on differentiable reward architectures for few-step diffusion models. By explicitly targeting non-differentiable signals—such as rigid OCR accuracy, binary human preferences, and discrete object counts—the research tackles a high-value practical constraint that currently limits the real-world deployment of ultra-fast generative models.



**2. Methodological Elegance & Variance Control:** The core architectural mechanism is technically astute. By leveraging the deterministic sampling trajectories inherent to Trajectory Distribution Matching (TDM), the authors successfully extract accurate, unbiased per-step reward estimates. Decoupling the optimization into dynamic surrogate reward learning (via group-based preference optimization) and generator fine-tuning effectively circumvents the severe variance explosion that typically cripples intermediate reward estimation in stochastic diffusion processes.



**3. Formidable Empirical Validation & Defense Against Reward Hacking:** The quantitative leap achieved by the proposed method is undeniable. Elevating a 4-NFE model's GenEval score from 61% to an exceptional 92%—thereby eclipsing both its 80-NFE base model (63%) and closed-source commercial benchmarks like GPT-4o (84%)—is a massive empirical achievement. Furthermore, the experimental design demonstrates maturity by proactively defending against the phenomenon of reward hacking. By validating the generative integrity through independent, out-of-domain visual quality metrics (e.g., Aesthetic Score, ImageReward, PickScore), the manuscript successfully proves that stringent instruction-following gains do not cannibalize fundamental visual fidelity.

### Weaknesses:

**1. Soundness (Theoretical): The Unquantified Bound** The derivation of the surrogate reward objective relies on a perilous mathematical shortcut. To achieve the tractable upper bound in Equation 8, Jensen's inequality is applied to force the expectation over $t$ and $q(x_t|x_{t_k})$ inside the sigmoid function. Optimizing a loose bound is structurally compromised, yet the manuscript completely fails to empirically quantify the tightness of this mathematical gap during training. If the bound diverges significantly, the surrogate is optimizing a phantom landscape.



**2. Soundness (Methodological): The Baseline Strawman** The primary ablation benchmarks TDM-R1 against a direct combination with an RL loss (DGPO). However, the text explicitly concedes that standard diffusion RL denoising losses are fundamentally incompatible with few-step models and inevitably produce blurry artifacts. Defeating a mathematically hobbled baseline that is guaranteed to fail does not scientifically validate the proposed architecture.



**3. Soundness (Analytical): The Performance Paradox** The manuscript reports that the 4-step generator (TDM-R1) consistently outperforms the 80-step model ($p_{\phi}$) that parameterizes its very own training signal. The justification for how a student transcends its teacher's ceiling relies entirely on a superficial, academically lazy analogy to LLM DPO literature. Continuous diffusion trajectories are not discrete text tokens; the paper lacks a rigorous mathematical proof explaining how the generator successfully filters out the base model's inherent generative limitations.



**4. Significance (Scope): Lack of Generalizability** The proposed mechanism is inextricably hardcoded to the specific deterministic trajectories of TDM. The paper provides zero evidence that this decoupled surrogate reward paradigm can generalize to other dominant few-step distillation architectures (e.g., Latent Consistency Models or Distribution Matching Distillation), artificially capping the broader impact of the theoretical contribution.

---

> ### Author Rebuttal · Authors · 2026-03-31
>
> We thank the reviewer for the comments. We would like to position that leveraging non-differentiable rewards to improve few-step generators at scales is one of the most important problems now for both academic and industry communities. To our best knowledge, TDM-R1 is the first attempt to systematically solve this problem at 6B parameter scales with significant improvements, where training with a single reward signal yields consistent OOD improvements across different backbones (SD3.5: Tab. 2, 6B ZImage: anonymous.4open.science/r/TDMR1-7881). We address each concern below.
> >The rationality of Jensen's inequality
>
> We clarify that using Jensen's inequality to derive a tractable training objective is consistent with established practice in Diffusion-DPO, which is a foundational and widely adopted method in this field. Jensen's inequality is routinely used across deep learning (e.g., the ELBO in VAEs). Diffusion-DPO has been extensively validated without explicitly tracking bound tightness, and this is standard practice in the community.
> >The Validity of the TDM w/ RL Loss Baseline
>
> We highlight that RL of few-step models with non-differentiable rewards is fundamentally an under-explored problem.
>
> The baselines suggested by the reviewer—DI* and Reward-Instruct—only support differentiable rewards and cannot handle non-differentiable rewards, making them inapplicable baselines for our setting.
>
> Among few-step methods formally supporting non-differentiable rewards, existing approaches are instances of combining DSM-based RL loss with few-step models. DSM loss is fundamentally incompatible with few-step models—the DSM optimal solution is the conditional expectation $E[x_0|x_t]$, leading to blurry few-step generation.
>
> "TDM w/ RL loss" baseline uses DGPO (SOTA, beating FlowGRPO), making it a strong representative of this class. Its failure reflects a fundamental limitation of existing works, not a "deliberately weakened" baseline choice. We demonstrate this both theoretically (DSM incompatibility) and empirically (Fig. 3 left, Fig. 4: progressive degradation under extended training). To our best knowledge, our work is the first systematic attempt of large-scale online RL for few-step models with non-differentiable rewards.
> >The Surrogate Performance Paradox
>
> We acknowledge that we currently lack a rigorous mathematical explanation for this phenomenon; the inherent complexity of few-step text-to-image generation makes such analysis challenging. Nonetheless, our empirical validation may serve as a useful foundation for future rigorous theoretical work.
>
> We provide further intuitive explanations: The surrogate reward $p_\phi$ generates images through 80 sequential denoising steps, suffering from error accumulation. The 4-step TDM-R1 generator is trained by directly supervising its trajectory, without suffering from error accumulation. Moreover, the generator benefits from marginal KL regularization, which effectively prevents reward hacking, yielding better OOD metrics.
>
> We highlight that this empirical finding is itself a valuable practical contribution: TDM-R1 can learn from an imperfect reward and exceed its generation performance. And it does not diminish our core contribution—the first successful systematic attempt of large-scale online RL for few-step models with non-differentiable rewards—which stands independently.
> >Scope of applicability
>
> **Conceptually, our method can reinforce any few-step model**. The use of deterministic few-step models is motivated by our theoretical analysis: deterministic trajectories enable low-variance per-step reward assignment. Importantly, deterministic (ODE-based) trajectories are the default sampling choice in both research and industry—FLUX, SD3, and most modern diffusions. Multiple few-step methods support deterministic trajectories, including the recently popular MeanFlow family. Our framework can be directly applied to any such architecture.
>
> **As shown in Fig. 3 (right), our method also works with stochastic trajectories**, i.e., employs DMD2 as the base few-step model with stochastic sampling. Our TDM-R1 can reasonably improve the performance of the variant, but it is worse than our default deterministic choice due to higher variance in per-step reward assignment — consistent with our theoretical analysis.
>
> We will expand this discussion in revision to clarify the scope of applicability.
> >Computational and memory Cost
> - Training TDM-R1 is efficient compared with standard Diffusion RL. Since the major cost of RL lies in rollout, TDM-R1 benefits from requiring only 4 NFEs for sampling. Training with GenEval takes <200 GPU hours, versus 2,000+ for Flow-GRPO and 200+ for DGPO—with matched performance.
> - We use LoRA for RL training, following standard practice. Since components are trained sequentially, the total memory overhead is essentially the additional LoRA parameters—negligible compared to the full model and gradient computation.
> Models|Memory
> -|-
> DGPO|64.5G
> Ours|68G

---

> > ### Author Rebuttal · Reviewer_axpY · 2026-04-03
> >
> > The rebuttal has been reviewed. The authors' clarifications regarding the baseline selection effectively address my core critiques. The practical impact of the work justifies publication, and I have adjusted my score upward.

---

> > > ### Author Response · Authors · 2026-04-04
> > >
> > > Thank you for acknowledging our work and for raising the score! Thank you again for your time and effort in reviewing our paper.

---

### Official Review · Reviewer_7Aum · 2026-03-07

**Soundness:** 4
**Presentation:** 3
**Significance:** 3
**Originality:** 3
**Overall Recommendation:** 5
**Confidence:** 5

**Summary:**

This paper proposes TDM-R1, a reinforcement learning framework designed for few-step diffusion models. Unlike existing RL approaches that rely on differentiable reward models, TDM-R1 enables the use of generic and non-differentiable rewards (e.g., human preferences or object counts). The method decouples training into surrogate reward learning and generator learning, and introduces a way to obtain per-step reward signals along the deterministic generation trajectory of TDM. Experiments on tasks such as text rendering, visual quality, and preference alignment show that TDM-R1 significantly improves few-step text-to-image models and achieves state-of-the-art RL performance on both in-domain and out-of-domain evaluations.

**Compliance With Llm Reviewing Policy:**

Affirmed.

**Final Justification:**

I thank the authors for the detailed rebuttal and the additional follow-up clarifications. The responses significantly improve my understanding of the paper, particularly regarding the core design choice and its distinction from existing diffusion RL methods.

**Soundness.** The method appears technically sound. The rebuttal clearly explains that the key mechanism is not merely the use of a surrogate reward, but the *decoupling* of reward learning and generator optimization, which removes the DSM-like objective from the generator update. The additional theoretical discussion (e.g., the one-step analysis showing the averaging/blurring effect of weighted DSM) and supporting experiments strengthen the technical justification. While some of these arguments could be presented more cleanly in the main paper, my main concerns about the correctness and rationale are largely resolved.

**Presentation.** The paper is reasonably structured, but some key insights were not sufficiently emphasized in the original version. The rebuttal and follow-up responses clarify the core idea much more effectively, especially the distinction between surrogate reward learning and generator learning. I encourage the authors to incorporate these explanations more explicitly into the main text to improve clarity.

**Significance.** The problem is important and timely, particularly for scaling RL with non-differentiable rewards to few-step generative models. The empirical results at 6B scale are strong and demonstrate clear advantages over existing approaches. The proposed framework has the potential to be practically useful for efficient generative modeling, especially in settings where reducing NFEs is critical.

**Originality.** I initially had concerns that the technical components (e.g., surrogate reward formulation) were closely related to prior work such as Diffusion-DPO. The rebuttal clarifies that the key novelty lies in the *decoupled optimization framework*, rather than the individual components themselves. This distinction is important: while parts of the method build on existing ideas, the way they are combined to address the DSM–few-step incompatibility is non-trivial. I therefore consider the level of originality to be acceptable.

Overall, the rebuttal has positively changed my evaluation. My main concerns regarding the necessity of a new method and its distinction from prior work have been largely addressed. I would still encourage the authors to better highlight the core insight and simplify the presentation in the final version, but the current contribution is sufficiently clear and meaningful.

Therefore, I support acceptance and will increase my score accordingly.

**Key Questions For Authors:**

1. What is the fundamental gap between reinforcing few-step and full-step diffusion models, and why does directly applying standard diffusion RL fail?
2. What specific challenges cause existing theoretically-grounded methods (Miao et al., 2025; Jiang et al., 2025) to not scale effectively?
3. What is the key advantage of TDM-R1 compared to these existing methods for non-differentiable rewards?
4. How does TDM-R1 specifically overcome the scaling challenges that previous methods face?

If the authors can provide a convincing explanation, I would be willing to reconsider my evaluation.

**Limitations:**

yes

**Strengths And Weaknesses:**

**Strengths**
1. This paper addresses the challenge of applying reinforcement learning to few-step diffusion models and enables fine-tuning with non-differentiable rewards.
2. The experimental results appear strong and demonstrate improvements across several evaluation settings.
3. The paper is clearly written and easy to follow, making the proposed method straightforward to understand.

**Weaknesses**

1. The motivation is not fully convincing to me. In the introduction, the authors argue that existing methods rely on differentiable rewards. However, the paper also acknowledges that several recent works have explored reinforcement fine-tuning of diffusion models using non-differentiable rewards. Therefore, a more compelling motivation would be to **analyze why existing RL methods for non-differentiable rewards fail to achieve satisfactory performance.** In my opinion, this issue represents the key challenge of the problem, yet the paper does not sufficiently discuss or even ignore it.
2. The paper lacks comparisons with several closely related methods. The authors state that “Some others theoretically support non-differentiable rewards (Miao et al., 2025; Jiang et al., 2025), but they have not validated their effectiveness at scale.” However, it is unclear (1) what exactly is meant by effectiveness at scale, and (2) why the proposed method succeeds in this setting while those methods cannot. Moreover, (3) the paper does not include empirical comparisons with these approaches. Since the experiments are primarily built upon the Flow-GRPO baseline, it is possible that the better (not reported yet) improvements over (Miao et al., 2025; Jiang et al., 2025) may partially stem from the underlying baseline rather than the proposed method itself.
3. The technical contribution appears somewhat limited. The proposed method seems largely based on a combination of existing techniques, particularly DMD and Flow-GRPO, without substantial new technical innovations. The real contribution of this paper appears to be adapting Flow-GRPO to the few-step generation setting and achieving improved results by combining it with DMD.

---

> ### Author Rebuttal · Authors · 2026-03-31
>
> We thank the reviewer for the comments. We would like to position that leveraging non-differentiable rewards to improve few-step generators at scales is one of the most important problems now for both academic and industry communities. To our best knowledge, TDM-R1 is the first attempt to systematically solve this problem at 6B parameter scales with significant improvements, where training with a single reward signal yields consistent OOD improvements across different backbones (SD3.5: Tab. 2, 6B ZImage: https://anonymous.4open.science/r/TDMR1-7881). We address each concern below.
> >Why Standard Diffusion RL fails for few-step models
>
> First, we clarify that few-step models are fundamentally different from standard diffusion models (DMs). Standard DMs are trained with the denoising score matching (DSM) loss to learn to gradually predict the conditional denoising ability via small discretization steps. Instead, few-step models use entirely different training objectives to learn to directly map between marginal distributions.
>
> Due to fundamental differences in learning mechanisms, directly applying diffusion RL on very few-step (e.g., 4-step) models will inevitably lead to unsatisfactory performance. The key is: existing diffusion RL methods supporting non-differentiable rewards can be reformulated as, or are equivalent to, weighted DSM loss (e.g., DDPO, Flow-GRPO, DGPO, AWM). The optimal $x_0$-prediction of DSM loss is $E[x_0|x_t]$, which is a blurry average over all possible clean images. Therefore, **DSM loss is incompatible with few-step and brutally using DSM-based RL on few-step models will make the few-step generation blurred**.
>
> "TDM w/ RL loss" baseline in Fig. 3 (left) and Fig. 4 clearly shows this: reward initially improves (benefiting from good few-step initialization) but degrades as training progresses—samples become increasingly blurry, exactly as theory predicts.
> >Why existing few-step RL methods fail and baseline fairness
>
> - **The few existing works that "theoretically" support non-differentiable rewards for few-step models are instances of directly combining DSM-based RL loss with few-step models, inheriting the fundamental DSM–few-step incompatibility** discussed above. Besides, they do not demonstrate online RL via non-differentiable rewards, validating only differentiable rewards. The performance of such methods would not continue to improve with scaling training compute—it would actually degrade due to the DSM–few-step incompatibility (as indicated by our "TDM w/ RL loss" experiments on this class of methods). This is what we mean by "not validated at scale."
> - "TDM w/ RL loss" baseline directly combines DGPO (SOTA Diffusion RL, beating FlowGRPO) with TDM, making it a faithful and even stronger instantiation of this class.  **Its failure demonstrates a fundamental limitation of existing works, not a weak baseline choice**. TDM-R1's significant advantage over this baseline stems from our fundamentally different approach of decoupling reward learning from generator learning, not from stronger underlying backbones.
>
> Sorry for not explicitly identifying existing methods as instances of "DSM-based RL + few-step model" in related work. We will clarify this in revision.
>
> > Clarify contribution
>
> We clarify our contributions as follows:
> - We position that leveraging non-differentiable rewards for few-step models is a fundamentally different, under-explored, yet important scientific problem. To our knowledge, TDM-R1 is the first systematic work that formulates and attempts to solve this problem.
> - The naïve approach (TDM w/ RL loss) fails due to the DSM–few-step incompatibility. Our key insight is that decoupling RL into surrogate reward learning and generator learning, such that the generator's loss contains no DSM component, is the critical enabler. Without this insight, the extension fails; with it, TDM-R1 matches SOTA standard diffusion RL performance with 20× fewer NFEs. Beyond that, specific technical contributions include: 1) extending DGPO from clean-sample to arbitrary noisy-latent reward learning for per-step optimization of few-step models; 2) marginal KL regularization (vs. conditional KL in FlowGRPO), avoiding unnecessarily strict constraints and achieving better OOD performance; 3) dynamic EMA reference model for improved stability and performance.
> - In this paper (as well as in the rebuttal phase), we successfully scale TDM-R1 at a scale of 6B parameters (based on Z-Image) and 4 generation steps. These scaling results bring novel impacts to the ultra-fast AIGC community.
>
> >Key advantage of TDM-R1 and how it overcomes scaling challenges
>
> TDM-R1 succeeds since its generator loss consists of: a) reward maximization via the surrogate reward and b) marginal KL regularization. Neither involves DSM loss. Since no DSM loss pushes the generator toward blurry, few-step quality is preserved and continues to improve with training (Fig. 3 left)—in stark contrast to "TDM w/ RL loss" which degrades.

---

> > ### Author Rebuttal · Reviewer_7Aum · 2026-04-01
> >
> > Although the authors addressed part of my concerns, for example an intuitive explanation of why existing approaches may not scale, several important questions still remain.
> >
> > - The paper argues that the main issue comes from a mismatch between the weighted DSM loss used in pretraining and the few-step generative model, and that a surrogate reward resolves this mismatch. I agree that, from a gradient view, DPO and GRPO can be interpreted as DSM-like objectives [1]. Still, I would like the authors to clearly explain **what the key design choice in the surrogate reward is that actually bridges this gap.** I also do not yet understand **why this same design cannot be applied on top of DPO or GRPO**, and why a new method is necessary.  It would help to **add experiments or analyses that directly support this claim**, and to expand the discussion on the limitations of the weighted DSM loss.
> >
> > - Because of the point above, I currently feel the insight and technical contribution may be somewhat limited. For example, **the core components of surrogate reward, such as Eq. 6, 7 and 8, are similar with the original Diffusion-DPO derivation [2]**.
> >
> > - That said, the problem is well motivated and the empirical results look strong. If the authors can (1) address the questions above in the next revision by adding the missing clarifications to the main text, and also (2) commit to releasing the full code and weights by a specified date, I would consider raising my score by 2 to 3 levels.
> >
> > [1] ConsistentRFT: Reducing Visual Hallucinations in Flow-based Reinforcement Fine-Tuning
> >
> > [2] Diffusion Model Alignment Using Direct Preference Optimization

---

> > > ### Author Response · Authors · 2026-04-03
> > >
> > > Thank you for the follow-up. We provide further clarifications below.
> > >
> > > We note that the surrogate reward does not work independently. The main point is that Eq. (6)–(8) are not the training objective of the few-step generator. The key design choice is that **the generator and surrogate reward are two different models**, which decouples the RL into two distinct phases: the differentiable per-step surrogate reward $\tilde r_\phi$ is trained by Eq. (6)–(8) with non-differentiable reward to estimate the reward of the student’s own noisy latents; in contrast, the few-step generator $g_θ$ is then optimized with Eq. (9)–(10): inspired by traditional RLHF (e.g., PPO), we assign a marginal reverse-KL term during the direct maximization of this differentiable surrogate reward. Specifically, the gradient of the expected reward $-E_{p_θ(x_{t_k})} \tilde r_{sg(\phi)}(x_{t_k})$ can be computed by $-E_{p_θ(x_{t_k})} \nabla_{x_{t_k}} \tilde r_{sg(\phi)}(x_{t_k}) \frac{\partial x_{t_k}}{\partial θ}$ (the surrogate reward is differentiable w.r.t. $x_{t_k}$).  This can be further simplified to $E_{p_\theta(x_{t_k})} E_{q(x_t|x_{t_k}), t\geq t_k} \gamma_t [s_{ref}(x_t) - s_\phi(x_t)] \frac{\partial x_{t_k}}{\partial θ}$, where $t$-related weighting is factored into $\gamma_t$.
> > >
> > > Therefore, the generator update does not involve DSM-like minimization. This direct maximization of the surrogate reward and the removal of weighted DSM from the generator update—not merely the existence of a surrogate reward—is the key mechanism for performing effective RL of few-step models.
> > >
> > > This also explains why our approach cannot simply be applied “on top of” DPO/GRPO. A DPO-like reward model can indeed be reused, and in fact we already do so in Eq. (6)–(8). However, if one also keeps the DPO/GRPO loss in the generator update, the few-step model still inherits the blurry property of the DSM objective. We provide additional experiments that optimize Eq. (6)-(10) with a single model (with results in https://anonymous.4open.science/r/TDMR1-7881/compare_tdmr1_single.pdf). It can be seen that the results are consistent with the "TDM w/ RL loss" baseline; the reward initially improves and then drops under extended training.
> > >
> > > Additionally, if one replaces that generator update with Eq. (9)–(10), then one has already changed the method into TDM-R1.
> > >
> > > **Further analysis of the failure of weighted DSM.**
> > >
> > > Let us consider the one-step case, which is the most challenging setting in practice but the most straightforward for theoretical analysis. Let $g$ be a one-step generator and consider a weighted DSM objective:  $E_{p(x)}E_{z \sim q(z|x) = N(0,I)} w(x) \lVert g(z)-x \lVert_2^2$.   In the one-step setting, $z$ is independent of $x$, so the optimal minimizer is $g(z) = E_{p(x)} w(x)x / E_{p(x)}w(x)$, which is a constant weighted average of samples within the model distribution, hence inherently blurry. This implies that under extended optimization, the objective has an averaging tendency that causes blur and degradation. As an additional sanity check, *we further perform experiments on reinforcing the **one-step** model* over SD-1.5 with HPSv2.1 as the training signal (results in https://anonymous.4open.science/r/TDMR1-7881/one_step_rl.pdf). The results are consistent with the "TDM w/ RL loss" baseline: both show early improvement followed by blurry outputs and later-stage degradation when a weighted-DSM-like loss is directly combined with few-step models.
> > >
> > > **Clarify surrogate reward learning**
> > >
> > > Our surrogate reward learning draws inspiration from DGPO (a group-level extension of Diffusion-DPO), but extends it in a significant way: rather than estimating the reward only for clean samples, our surrogate reward estimates the reward of arbitrary noisy latents along the student trajectory, thereby better supporting per-step learning for few-step generators. This extension yields notable performance gains, as shown in Fig. 6 Left, where TDM-R1 outperforms a variant that relies on a frozen, well-trained DGPO as the surrogate reward.
> > >
> > > To our knowledge, leveraging models trained via Diffusion-DPO-like objectives as surrogate rewards for reinforcing a separate generative model has not been previously explored. This enables a natural decoupling of the RL process into two distinct phases: surrogate reward learning and generator learning.
> > >
> > > This decoupled design enables joint learning of the DPO-style reward model and the generator, serving as a universal RL framework for few-step and even one-step generation with native support for non-differentiable rewards. Whereas naively applying standard diffusion-based RL to few-step generation typically yields blurry outputs, **our approach provides a substantially more practical pathway for bringing RL to one- and few-step models**. We believe this represents a valuable contribution to the community.
> > >
> > > **Open-source Commitment**
> > >
> > > We will release our training code and pre-trained weights no later than the publication date.

---

### Official Review · Reviewer_ohdv · 2026-03-18

**Soundness:** 3
**Presentation:** 1
**Significance:** 3
**Originality:** 2
**Overall Recommendation:** 4
**Confidence:** 5

**Summary:**

The paper tackles the task of RL post-training a few-step generative text-to-image model using non-differentiable rewards. The authors do this by post-training a diffusion model with DGPO, converting it into a surrogate reward used to fine-tune the few-step generator, and training both models jointly. Experimentally, they evaluate the method on GenEval and text-rendering benchmarks and show impressive performance that is on par with prior DiffusionRL methods, along with several ablations on the different design choices.

**Compliance With Llm Reviewing Policy:**

Affirmed.

**Final Justification:**

This paper is a timely one that discusses how to RL-finetune an already distilled few-step generator based on diffusions. My concerns were primarily with the writing quality and the lack of explanations/clarifications in certain aspects of the paper. The authors have mostly clarified these points and have promised to update the final paper with these clarifications. As such, I believe this to be a good submission and I recommend acceptance.

**Key Questions For Authors:**

- Based on my understanding of the approach, the diffusion model used to obtain the surrogate reward is itself trained with DGPO. However, in Table 2, comparing lines 311 and 319, the DGPO baseline appears to outperform it. What could explain this performance gap?
- Prior work, such as DiffusionNFT, has found the EMA decay rate to be an important hyperparameter for achieving strong performance. How was `ema_decay` set in your experiments and did you experience similar sensitivity to this parameter?

**Limitations:**

Yes.

**Strengths And Weaknesses:**

Strengths:

- The core premise of the paper is interesting, important, and quite timely. A great deal of recent work has studied RL post-training for diffusion models, but relatively little has extended these methods to few-step models. As data modalities grow larger, from images to video, and models become heavier and more complex, few-step models are becoming increasingly relevant, making the ability to post-train them increasingly important.
- The experimental results are promising.

Weaknesses:

- The writing can be very confusing at times. For example:
    - Section 3.1 discusses how to assign reward values to intermediate noisy latents $x_t$. In lines 143–148, the authors cite prior work and state that assigning the reward value of the final clean sample to the entire trajectory can introduce bias into the rewards assigned to intermediate steps. A few paragraphs later, in lines 168–173, they argue that by doing something very similar, they can obtain unbiased reward estimates. This is unclear and needs further clarification.
    - The symbols $\theta, \phi, \psi, \dots$ are mixed together frequently and at times used interchangeably, which makes it difficult to understand which model each symbol refers to. For example, comparing the equation on line 216 with Eq. 6, the right-hand side uses both $p_\theta$ and $p_\phi$.
    - Lines 247–252 discuss the difference between the KL regularization in Eq. (9) and the KL regularization used in prior work such as FlowGRPO. As far as I understand it, FlowGRPO also includes an identical KL minimization term, but evaluated at $t=0$ on clean data, whereas here it is evaluated at multiple intermediate timesteps $t_k$. Since both minimize a KL divergence between two distributions, in what sense are they different? This needs further clarification.
    - Lines 336–345 discuss the interesting phenomenon that the fine-tuned diffusion model $p_\phi$ performs worse than the few-step student TDM-R1. This is quite interesting, but the explanation provided is somewhat strange. The authors refer to the LLM literature, where it has been observed that “improving the model’s performance as a reward signal does not necessarily translate to improved generation performance.” This does not seem directly applicable here, since the diffusion model $p_\phi$ is itself being trained with DGPO, a post-training method intended to improve inference-time performance. This point needs more clarification.
- Related to the previous point, the paper would greatly benefit from an overview figure or pseudocode for the final algorithm. Adding either would go a long way toward helping readers understand the approach more easily.
- Missing citations: the “Dynamic Reference Model” section incorrectly assumes that all prior work updates the sampling policy $\theta_{\text{old}} / \theta_\text{ref}$ to the current policy at fixed intervals. However, using an EMA has been explored before, for example in DiffusionNFT.

---

> ### Author Rebuttal · Authors · 2026-03-31
>
> We thank the reviewer for the comments. We would like to position that leveraging non-differentiable rewards to improve few-step generators at scales is one of the most important problems now for both academic and industry communities. To our best knowledge, TDM-R1 is the first attempt to systematically solve this problem at 6B parameter scales with significant improvements, where training with a single reward signal yields consistent OOD improvements across different backbones (SD3.5: Tab. 2, 6B ZImage: https://anonymous.4open.science/r/TDMR1-7881). We address each concern below.
> > Lines 143–148 vs. 168–173: reward estimation contradiction
>
> Sorry for confusion. "introduce bias" in lines 143–148 is a typo, which should read "introduce high variance".  Both cases are unbiased estimates of  $r(x_t, c)\triangleq E_{p(x|x_t)}[r(x, c)]$, but differ in variance.
> The key is: the estimation variance depends on the stochasticity of $p(x|x_t)$.
>
> Intuitively, under stochastic trajectories, computing $r(x_t)$ requires taking an expectation, which inevitably introduces variance; however, under deterministic trajectories, the expectation reduces to a direct computation, introducing no estimation variance.
> This is a valuable finding of our work: it provides a principled theoretical justification for choosing deterministic few-step models (e.g., TDM) for RL post-training, rather than heuristically applying RL to arbitrary few-step models.
> >Symbol confusion
>
> In general, $θ$ denotes the generator, $\phi$ denotes the surrogate reward, and $ψ$ denotes the pre-trained score. The use of $p_θ$ on the RHS of Eq. 6 is a typo. We will fix all inconsistencies and add a symbol reference table.
> >KL term vs. Flow-GRPO
>
> The difference is fundamental and, moreover, leads to better OOD metrics.
> FlowGRPO uses a **conditional** KL:
> $$KL(p_\theta(x_{t-1}|x_t)||p(x_{t-1}|x_t)),$$
> requiring the model's prediction at every point along the trajectory to match the reference model.
> TDM-R1 uses a **marginal** KL:
> $$KL(p_{θ}(x_t)||p(x_t)),$$
> only requiring the model's overall distribution to match the reference. For RL post-training, regularization aims to keep the model's outputs within the image manifold—we do not need the denoising trajectory to mimic the reference. FlowGRPO's conditional KL imposes a difficult constraint, while **marginal KL leads to a simpler yet effective regularization**, which benefits the RL. It introduces sufficient regularization to prevent reward hacking without notably slowing training, resulting in better OOD metrics with only 4 NFEs than 80-NFE FlowGRPO (Tab. 2).
> >$p_\phi$'s generation quality: underperforming DGPO
>
> We appreciate your novel intuition. We do believe it to be likely that some discriminative reward is more suited for few-step model RL (Fig. 6 left: TDM-R1 beats TDM-R1 w/ frozen DGPO as reward), but not necessarily corresponds to a better generative diffusion model. Actually, the surrogate reward in TDM-R1 is tailored for per-step reward learning, which may deviate from the one in DGPO due to two factors:
> - Train-eval mismatch. The surrogate reward is optimized for reward estimation at every $x_t$ along the student's trajectory, not reward capacity and generation quality only at $x_0$.
> - Distribution shift. DGPO trains on its own rollouts, while our surrogate trains on few-step model's rollout, creating a hybrid on/off-policy regime—consistent with the widely observed finding that offline DPO learns good rewards (i.e., good discrimination between positive and negative samples) while underperforming online DPO in generation. The DGPO paper also observed that its online version shows better performance (DGPO's Fig. 5).
>
> >Add Pseudocode
>
> Great suggestion. We will add pseudocode in the revision.
> >Discuss DiffusionNFT‘s EMA and decay sensitivity
>
> We will add the discussion in the revision. Notable differences exist in how EMA is used:
> - NFT's EMA "old" model serves as the rollout policy, coupled with sampling; our EMA reference model is part of the reward parameterization and is decoupled from sampling.
> - NFT's EMA influences the gradient toward the old policy (simplified gradient: $2Aβ(v_{old} - v_{gt}) + 2β^2(v_\phi - v_{old})$), so large EMA decay directly slows training. Our EMA reference model only modulates regularization strength without directly affecting gradient direction in reward learning (simplified gradient: $σ(Aβ(T-t_k)∑(KL_\phi-KL_{ref}))).detach()*Aβ(T-t_k)\nabla KL_\phi$). This makes our method substantially less sensitive. We set EMA decay to linearly increase from 0.1 to max 0.5. As Fig. 6 shows, even a fully frozen reference (decay=1) achieves reasonable performance, confirming robustness. To our knowledge, using EMA-updated references in a diffusion DPO-like framework has not been previously explored.

---

> > ### Author Rebuttal · Reviewer_ohdv · 2026-04-03
> >
> > I appreciate the authors' response. My concerns are mostly addressed and as a result, I am raising my score.

---

> > > ### Author Response · Authors · 2026-04-04
> > >
> > > Thank you for your acknowledgment of our work and for raising the score! Thank you again for your time and effort in reviewing our paper.

---

### Official Review · Reviewer_K7sK · 2026-03-26

**Soundness:** 1
**Presentation:** 3
**Significance:** 4
**Originality:** 2
**Overall Recommendation:** 5
**Confidence:** 4

**Summary:**

This paper proposes TDM-R1, a reward fine-tuning framework for few-step diffusion models that learns step-wise surrogate rewards along TDM trajectories and uses group-based preference optimization to improve generation quality. It argues that standard diffusion RL is poorly matched to the few-step setting, and instead introduces a more direct reward-learning approach tailored to fast generation. Empirically, the method shows strong gains on compositional generation, text rendering, and human preference alignment.

**Compliance With Llm Reviewing Policy:**

Affirmed.

**Key Questions For Authors:**

-

**Limitations:**

-

**Strengths And Weaknesses:**

I lean toward **clear accept** for this paper. Reward fine-tuning for few-step diffusion models is a highly timely topic, and although I do not view most of the individual ingredients of the paper as fundamentally novel in isolation, I believe the paper makes a meaningful contribution by integrating several previously fragmented components into a coherent framework. In particular, combining TDM-style few-step generation with group-based reward learning is a highly plausible design choice, and the fact that this combination leads to strong empirical gains is, in my view, an important finding worthy of publication.

That said, I believe the paper has several conceptual and theoretical issues that should be clarified:

1. **Interpretation of $r(x,c)$ as a probability.**
   The paper appears to use $r(x,c)$ directly in the form of a probability, which I find conceptually questionable. At the very least, I would have expected some effort to introduce an explicit normalization term, or alternatively to define a probability model of the form $\exp(-r(x,c))$ followed by normalization. As written, the probabilistic interpretation seems under-justified. Is there a technical reason why the derivation would fail under a more principled probabilistic modeling choice?

2. **Claim of unbiased estimation in Eq. (5).**
   I also find the wording around statements such as *“This equation (5) actually justifies the reasonableness of defining the reward for the entire diffusion path using the reward of $x_0$”* and *“In other words, if $p(x \mid x_t)$ is a deterministic Dirac distribution”* problematic. In principle, approximating the SDE posterior using an ODE-style trajectory may be a reasonable practical approximation. However, in that case, I believe the paper should state this much more explicitly and honestly: the target object is still the SDE posterior, but since exact posterior sampling is intractable, the method approximates it via rollout from a learned few-step ODE-like model and then uses this sample as a Monte Carlo estimator. Such an estimator is generally **biased**, not unbiased as claimed in the paper. I therefore do not think the current “unbiased estimation” claim is justified.

3. **Unclear justification for treating TDM as learning an ODE trajectory.**
   More fundamentally, I do not understand why TDM should be interpreted as learning an ODE trajectory in the first place. From my reading, TDM applies a distribution/distillation matching objective not only for the mapping $t \to 0$, but also for intermediate mappings such as $t \to u$. However, it is not clear to me why this alone is sufficient to justify the claim that the model learns an ODE trajectory. If such a justification does not actually hold, then I believe a substantial part of the conceptual foundation of TDM-R1 becomes weakened as well.

Overall, I find the paper interesting, timely, and empirically valuable, and I am positive on acceptance. However, I believe the theoretical narrative should be made substantially more precise, especially regarding the probabilistic interpretation of reward, the estimation bias in Eq. (5), and the role of TDM as an alleged ODE-based trajectory model.

---

> ### Author Rebuttal · Authors · 2026-03-31
>
> We thank the reviewer for the comments. We would like to position that leveraging non-differentiable rewards to improve few-step generators at scales is one of the most important problems now for both academic and industry communities. To our best knowledge, TDM-R1 is the first attempt to systematically solve this problem at 6B parameter scales with significant improvements, where training with a single reward signal yields consistent OOD improvements across different backbones (SD3.5: Tab. 2, 6B ZImage: https://anonymous.4open.science/r/TDMR1-7881). We address each concern below.
>
> > Interpretation of $r(x, c)$ as a probability.
>
> Great question.  We clarify that our probabilistic interpretation of $r(x,c)$ is indeed normalized. We interpret the $r(x,c)$ that is normalized to [0,1] as the parameter of a Bernoulli distribution:
> $$P(\text{good} \mid x, c) = r(x, c), \quad P(\text{not good} \mid x, c) = 1 - r(x, c).$$
> Since $(r(x,c) + (1-r(x,c)) = 1$, the probability is normalized by construction.
>
> > Claim of unbiased estimation in Eq. (5).
>
> We first provide an intuitive explanation under the SDE diffusion formulation to facilitate discussion: from $x_t$, we can traverse infinitely many paths to reach $x_0$, then evaluate the reward and take the average to obtain $r(x_t)$. Under the few-step deterministic path setting, each xt corresponds to a unique $x_0$, so we only need to follow a single path from xt to reach x0, then evaluate the reward to obtain $r(x_t)$.
>
> From a high-level conceptual perspective, this $r(x_t)$ is a modeling choice, jointly determined by $p(x_0 \mid x_t)$ and $r(x_0)$:
> $$r(x_t, c) \triangleq \mathbb{E}_{p(x|x_t)}[r(x, c)].$$
> In practice, **different choices of $p(x_0 \mid x_t)$ yield different $r(x_t, c)$**. Both cases are unbiased, but differ in variance.
> When we choose a deterministic $p(x|x_t)$ — as is the case for TDM's deterministic trajectories—then it can be computed accurately via a single-point evaluation with zero variance. In contrast, choosing a stochastic $p(x|x_t)$ would define a different $r(x_t, c)$, and a single-sample estimate would have nonzero variance. We will revise the text to clearly frame $p(x|x_t)$ as a design choice that jointly models the intermediate reward.
>
> > Unclear justification for treating TDM as learning an ODE trajectory.
>
> Insightful observation. The design intuition of TDM is motivated by the effectiveness of ODE sampling. However, after training, TDM only learns transitions between two marginals, which do not actually correspond to solving an ODE. Instead, it learns to perform few-step sampling by transitioning between different marginals through deterministic mappings.
>
> In other words, TDM learns a deterministic trajectory from noise to images, which shares the key structural property of ODE trajectories: given $x_t$, the output $x_u$ is uniquely determined. Our analysis regarding the per-step reward assignment requires only that $p(x|x_t)$ is a Dirac delta distribution—i.e., that the trajectory is deterministic — and TDM satisfies this requirement. We will add a discussion regarding this in the revision to make it clearer.

---

> > ### Author Rebuttal · Reviewer_K7sK · 2026-04-05
> >
> > Thanks for the response. It resolved my questions and I'll keep my score.

---

### Decision · Program_Chairs · 2026-04-30

**Decision:**

Accept (regular)

**Comment:**

TDM-R1 introduces an RL post-training framework for few-step diffusion models that supports non-differentiable rewards, a setting where existing methods fail due to the fundamental incompatibility between denoising score matching (DSM) objectives and few-step generation. The key insight is decoupling RL into surrogate reward learning and generator optimization, ensuring the generator update contains no DSM component. Scores are 5/5/4/4, with all reviewers positive after rebuttal.

Strengths:
  - Addresses a timely and important problem. This is the first systematic attempt at large-scale online RL for few-step models with non-differentiable rewards, validated at 6B scale.
  - Strong empirical results: GenEval improves from 61% to 92% on a 4-step model, exceeding the 80-step base model and matching GPT-4o, with preserved out-of-domain metrics.
  - The theoretical argument for why DSM-based RL degrades few-step models is clearly articulated and empirically validated through the "TDM w/ RL loss" ablation.

All four reviewers' concerns were substantially addressed during rebuttal. Reviewer K7sK's theoretical questions (probabilistic interpretation, ODE trajectory justification) were clarified. Reviewer 7Aum's novelty concern was resolved through additional experiments showing that single-model optimization degrades like the DSM baseline, confirming the decoupled design is the core contribution. Reviewers ohdv and axpY's concerns about presentation and theoretical tightness were acknowledged, with fixes promised for camera-ready.

The Jensen bound remains unquantified and the student-surpassing-teacher phenomenon lacks formal proof. These should be explicitly acknowledged as limitations. The authors should incorporate rebuttal clarifications (notation, pseudocode, expanded related work) into the camera-ready.